

# Influence of microorganisms on initial soil formation along a glacier forefield on King George Island, maritime Antarctica

Patryk Krauze[1], Dirk Wagner[1,2], Diogo Noses Spinola[3,4] and Peter Kühn[3]

[1]GFZ, German Research Centre for Geosciences, Helmholtz Centre Potsdam, Section Geomicrobiology, 14473 Potsdam, Germany

[2]Institute of Geosciences, University of Potsdam, 14476 Potsdam, Germany

[3]Department of Geosciences, Research Area Geography, Laboratory of Soil Science and Geoecology, Eberhard Karls University Tübingen, 72070 Tübingen, Germany

[4]Present address: Department of Chemistry and Biochemistry, University of Alaska Fairbanks, 99775-6160 Fairbanks, USA

*Correspondence to*: Patryk Krauze (pkrauze@gfz-potsdam.de)





**Abstract**. Compared to the 1970s, the edge of the Ecology Glacier on King George Island, maritime Antarctica,
is positioned more than 500 m inwards, exposing a large area of new terrain to soil-forming processes and
periglacial climate for more than 40 years. To gain information on the state of soil formation and its interplay
with microbial activity, three hyperskeletic Cryosols (vegetation cover of 0 – 80 %) in the recently (< 50 years)
deglaciated foreland of the Ecology Glacier and a Cambic Cryosol (vegetation cover of 100 %) behind a lateral
moraine deglaciated more than 100 years ago were investigated by combining soil chemical and microbiological
methods. All soils are formed in the same substrate and have a similar topographic position. In the upper part of
all soils, a decrease in soil pH was observed, but only the Cambic Cryosol showed a clear direction of pedogenic
and weathering processes. Differences in the development of these initial soils could be related to different
microbial community composition and vegetation coverage, despite the short distance among them. We observed
- decreasing with depth - the highest bacterial abundances and microbial diversity at vegetated sites. All soils
were dominated by bacterial phyla such as Proteobacteria, Actinobacteria, Bacteroidetes, Acidobacteria,
Verrucomicrobia, and Chloroflexi. Multiple clusters of abundant OTUs were found depending on the site-
specific characteristics as well as a distinct shift in the microbial community structure towards more similar
communities at soil depths > 10 cm. In the foreland of the Ecology Glacier, the main soil-forming processes on a
decadal timescale are acidification and accumulation of soil organic carbon and nitrogen, accompanied by
changes in microbial abundances, microbial community compositions, and plant coverage, whereas quantifiable
silicate weathering and the formation of pedogenic oxides occur on a centennial to a millennial timescale after
deglaciation.



## 1 Introduction

Retreating glaciers in polar and mountainous regions reveal proglacial terrain that is exposed to soil formation and subsequently colonized by microorganisms and plants (Matthews, 1992; Walker and del Moral, 2003; Mavris et al., 2010; Bradley et al., 2014). Considering the particular vulnerability of the Antarctic environment to climate change, studies on soils from glacier forelands could provide indications of how climate changes at the global scale will affect soil formation at the regional scale. By substituting space with time, chronosequences of proglacial environments are an important tool to understand primary succession and soil forming processes (Walker et al., 2010), and were therefore used to study the succession of soil microbial communities and their influence on initial soil formation in the past (e.g. Nemergut et al., 2007; Schmidt et al., 2008, Wojcik et al., 2018). Such microbial populations with different abundances, community structures and diversities are among the first organisms to colonize recently deglaciated areas (Sigler and Zeyer, 2002; Bajerski and Wagner, 2013; Rime et al., 2015). Their activities within biogeochemical cycles such as the fixation of carbon and nitrogen into bioavailable forms (Nemergut et al., 2007; Schmidt et al., 2008) can promote environmental changes that facilitate the succession of organisms at higher trophic levels (Smith, 1993; Hämmerli et al., 2007; Donhauser and Frey, 2018). In order to understand the relationship between primary and secondary succession in proglacial environments and to shed light on the influence of microbial processes on the development of initial soil ecosystems and vice-versa, it is crucial to study the factors that shape the genetic structure of local microbial populations of such environments (Hämmerli et al., 2007).

The present microbial communities in ice-free areas of polar regions are dominated by Acidobacteria, Actinobacteria, Bacteroidetes, Firmicutes, Gemmatimonadetes and Proteobacteria (Chong et al., 2012; Bajerski and Wagner, 2013; Ganzert et al., 2014), which are well adapted to their harsh environment (Bajerski et al., 2017; Mangelsdorf et al., 2017). These microbial communities have been described to be influenced by local soil chemical parameters, such as pH (Siciliano et al., 2014), and soil physical parameters such as grain size distribution and soil moisture (Ganzert et al., 2011). Also, the microclimate (Cannone et al., 2008), vegetation cover, and cryoturbation processes (Almeida et al., 2014) play a role in the observed soil properties (e.g. bulk density, soil temperature) in Antarctica. Thus, these properties and processes could have an impact on soil microbial community composition and activity.

Compared to ice-free areas of continental Antarctica, the soils from maritime Antarctica differ significantly due to higher water availability and warmer temperatures, which lead to deeper active layers and promote vegetative cover and mineral weathering (Campbell and Claridge, 1987; Blume et al., 2004; Ugolini and Bockheim, 2008).





Following the regional warming during the last 50 years, a significant loss of ice volume and melting of many
outlet glaciers in maritime Antarctica could be observed (Braun and Gossmann, 2002; Cook et al., 2005; Simoes
et al., 2015). The glacial retreat will probably keep its accelerated pace due to the continuous warming over
Antarctica by 0.34 °C per decade (Turner et al., 2014). It will continuously affect soil-forming processes and
microbial activity in maritime Antarctica by exposing finely textured glacial sediments (Strauss et al., 2012;
Bockheim et al., 2013; Vlček, 2016), which offers an excellent setting to investigate initial soil-forming
processes and the colonization by microbial pioneers before higher plants succeed (Strauss et al., 2012; Bradley
et al., 2014). Particularly, the frontal retreat of the Ecology Glacier on King George Island (KGI), South
Shetland Islands, with approx. 30 m per year since the early nineties opens new terrain for soil-forming
processes and terrestrial life (Birkenmajer, 2002). In a centennial to millennial timescale, the carbon and nitrogen
content, as well as the pH are the main soil properties to change on KGI, leading to the formation of soil
horizons (Boy et al., 2016). Similar findings were observed from glacier forelands in Europe, particularly from
the Alps in Switzerland (Dümig et al., 2011; Mavris et al., 2011; Dümig et al., 2012; Mavris et al., 2012). A
recent study in the foreland of the Ecology Glacier on KGI demonstrated that the diversity and properties of
microorganisms in recently deglaciated areas are not only related to age but also to differences in soil stability
within the upper centimeters due to the influence of cryoturbation (Zdanowski et al., 2013). Nevertheless, there
is still a deficiency of information about the decadal-scale changes of soil properties and their interplay with
microorganisms in soil ecosystems in maritime Antarctica.
The objective of this study was to identify the main initial pedogenic processes in relation to microbial
community structure and microbial abundances in the recently (< 50 years) deglaciated foreland of the Ecology
Glacier. To keep the soil formation factors constant, three soils were sampled at a distance of 150 m formed on
the same substrate and in a similar topographic position, but with different vegetation cover. These soils were
compared with an older soil formed on a similar substrate that had been deglaciated for over 100 years. We
combined grain size and pedochemical analyses with DNA-based molecular biological analyses, including high-
throughput sequencing and quantitative PCR, to determine the diversity, distribution, and abundance of
microbial communities.

**2 Material and Methods**
**2.1 Study Area**



King George Island is located in the South Shetland Islands archipelago. The stratigraphy of KGI comprises
Upper Cretaceous to Lower Miocene predominantly subaerial volcanic and volcanoclastic rocks. Fossiliferous
marine and glaciomarine sediments are more common in the Oligocene to Lower Miocene rocks. Quaternary
volcanism along the southern margin of KGI and the axial part of Bransfield Strait is related to back-arc
extension (Birkenmajer, 1980). The rocks exposed after the frontal retreat of the glacier are mainly mafic
volcanic rocks from the Arctowski Cove Formation.
The relatively mild and moist conditions in maritime Antarctica compared to continental Antarctica result in
frequent freeze-thaw cycles, which foster periglacial processes (e.g. cryoturbation), and chemical and physical
weathering (Simas et al., 2008). Moreover, a usually water-saturated active layer during the summer increases
biological and chemical weathering and accumulation of organic carbon (Bockheim, 2015). As a result, a suite of
soil-forming processes occurs on the island, such as cryoturbation, gleization, melanization, paludization, and
phosphatization. Since most of the soils are relatively young and weakly developed (~ 4000 yr BP since the last
deglaciation on KGI; Yoon et al., 2000), these processes are closely linked to the landscape position, parent
material and faunal activity (e.g. penguin rookeries). The main resulting soil orders are Cryosols, Leptosols,
Cambisols, and Histosols (Simas et al., 2007; Simas et al., 2008; Bockheim, 2015), but also different soil groups
such as Arenosols and Gleysols may occur (Michel et al., 2014). Additionally, Podzols, Umbrisols, Stagnosols
and Gleysols were found in the surrounding area of the Arctowski Station on King Georges Island (Bölter et al.,
1997; Blume et al., 2002b).
The study site is located in the foreland of the Ecology Glacier on KGI characterized by an oceanic polar
climate. Temperature measurements recorded by the Chilean Antarctic station President Eduardo Frei Montalva
from 1971 to 2004 indicate a mean annual temperature of -2.3 °C with the coldest temperatures in July and
August (mean temperature -6.5 °C) and the warmest in February (mean temperature 1.6 °C). The mean annual
precipitation is < 500 mm, with maximum precipitation during spring/autumn and a minimum during
summer/winter (Cerda, 2006). The margin of the Ecology Glacier was > 500 m inward in 2014 compared with
the front line during the late 1970s, where the ice reached the sea (Figure 1). The present coastline represents
approximately the front line of summer 1956/1957 after Birkenmajer (2002).
**[Figure 1]**
The profiles KGI A, B, and C are located within 150 m distance on a substrate deglaciated for around 50 years.
These sites are within the sampling zone III of Zdanowski et al. (2013). We sampled three soil profiles (A, B, C)



from well-drained positions of lateral moraine deposits with slightly different leeside/windward positions in the
present foreland of the Ecology Glacier (Figure 2).
**[Figure 2]**
One soil profile (KGI D) is directly located beyond the lateral moraine (Figure 1) on a substrate > 100 years old.
The substrate of all profiles is mainly composed of volcanic material from the Arctowski Cove Formation. None
of the investigated soil profiles are influenced by penguin or bird rookeries.
Fieldwork was carried out in summer 2014. Soil morphological description followed the guidelines of the Food
and Agriculture Organization of the United Nations (FAO, 2006), and the pedons were classified using the
World Reference Base system (WRB, 2015). Samples were transported frozen to Germany and stored at a
temperature of -18 °C.
**2.2 Soil Physics**
Volumetric samples (100 cm$^3$) for bulk density were taken from each horizon/depth increment with steel rings in
three replicates. Bulk density [g cm$^{-3}$] was gravimetrically determined including the correction by coarse
material (Eq. 1; Henkner et al., 2016). Bulk samples were air-dried and sieved < 2 mm. The grain size
distribution (< 2 mm) of all samples was determined by combined sieving (2000 µm to 20 µm) and X-ray
granulometry after using 1 M sodium metaphosphate (Na$_4$P$_2$O$_7$) as a dispersant (Blume et al., 2011).
**2.3 Pedochemical analyses and calculation of pedogenic oxide ratios and the Chemical Index of Alteration**
Total nitrogen (N$_t$) and soil organic carbon (SOC) were determined by thermal conductivity analysis after heat
combustion (1150 °C) with a CNS-element analyzer (Elementar Vario EL III). Soil pH$_{[H2O]}$ and pH$_{[CaCl2]}$ were
determined potentiometrically in a 1:2.5 soil to water/0.01M CaCl$_2$ solution. Pedogenic Fe-(hydr-)oxides (Fe$_d$)
were extracted by dithionite-citrate-bicarbonate (DCB) solution (Mehra and Jackson, 1960). Non- and poorly
crystallized compounds of Fe (Fe$_o$) and Al (Al$_o$) were extracted by shaking 2.5 g of soil in 100 mL 0.2 M acid
ammonium oxalate (pH 3) for 4 h in the dark (Schwertmann, 1964). The ratio between total Fe and pedogenic
Fe-(hydr-)oxides (Fe$_t$/Fe$_d$) gives information on the iron release of Fe-bearing minerals, reflecting the intensity of
weathering, whereas the ratio Fe$_o$/Fe$_d$ gives information on the degree of iron oxides crystallinity (Arduino et al.,
1986). Major elements, including Fe (Fe$_t$), were measured with a wavelength dispersive XRF device
(PANanalytical PW 2400). Prior to preparation, the bulk samples (ratio Li-metaborate to soil 1:5) were ground
with an agate mill for 10 minutes. The Chemical Index of Alteration (CIA) gives information on the ongoing



chemical weathering and was calculated according to Nesbitt and Young (1982). The calculation was as follows
$[(Al_2O_3 / Al_2O_3 + Na_2O + CaO* + K_2O)) \times 100]$, where $CaO*$ represents the amount of silicate-bound CaO.
**2.4 Nucleic acids extraction**
The total genomic DNA of each sample was extracted in triplicates with the FastDNA™ Spin Kit for soil (MO
BIO Laboratories Inc., USA). Samples with very low DNA yields were extracted three times and the extracts
were pooled. DNA extracts were stored at -20 °C and used as templates in the quantification the bacterial 16S
rRNA gene and high-throughput (HiSeq) sequencing.
**2.5 Illumina HiSeq-Sequencing**
Total genomic DNA extracts of each sample were sequenced using tagged 515F (5'-
GTGCCAGCMGCCGCGGTAA-3') and 806R (5'-GGACTACHVGGGTWTCTAAT-3') primers after
Caporaso et al. (2010). The used cycler program and reaction mix were described by Meier et al. (2019). The
sequencing was performed on an Illumina HiSeq (2 x 300 bp) by GATC Biotech AG, Germany.
**2.6 Bioinformatics and statistical analysis**
Raw sequencing data obtained by Illumina HiSeq (2 x 300 bp) was processed according to Meier et al. (2019)
with few differences. Clustering of operational taxonomic units (OTUs) at 97 % sequence similarity and their
taxonomic assignments was done with the SILVA data base (version 132, Quast et al., 2013). Resulting data
were visualized using R and PAST3 (Hammer et al., 2001). Demultiplexed raw sequencing data were deposited
at the European Nucleotide Archive (http://www.ebi.ac.uk/ena) under the accession number PRJEB37594.
**2.7 Quantification of bacterial 16S rRNA gene copy numbers**
Bacterial abundances were quantified using quantitative PCR (qPCR) and the 314F (5'-
CCTACGGGAGGCAGCAG-3') and 534R (5'-ATTACCGCGGCTGCTGG-3') primers after Muyzer et al.
(1993). The used cycler program and reaction mix were described by Meier et al. (2019).
**3 Results**
**3.1 Soil classification and soil properties**



The approximately 50-year-old soils KGI A, KGI B, and KGI C did not have properties to differentiate soil
horizons and were classified as Hyperskeletic Cryosols. The older soil KGI D had distinct soil horizons and was
classified as a Cambic Cryosol. Differences in vegetative cover and pedochemical properties were observed
between the investigated soil profiles (Table 1).
**[Table 1]**
Regarding soil pH, similar trends were observed in the investigated soil profiles. The lowest $pH_{H20}/pH_{CaCl2}$ was
found in the uppermost depth increment (A: 7.9/6.7; B: 7.4/6.5; C: 6.1/5.3; D: 5.1/4.8). With depth,
$pH_{H20}/pH_{CaCl2}$ increased with the highest values in the lowermost depth increment (A: 8.8/7.7; B: 8.5/7.4; C:
8.05/6.99; D: 7.5/6.3). The vegetation cover was 0 %, 5 %, 80 %, and 100 % for KGI A, KGI B, KGI C, and
KGI D, respectively. The vegetation cover at KGI B included *Usnea antarctica*, *Deschampsia antarctica*, and
*Colobanthus quitensis*. In addition, no significant accumulation of nitrogen ($N_t < 0.03$ %) nor soil organic carbon
(SOC < 0.05 %) was observed. KGI C had higher $N_t$ (0.09 %) and SOC (1.24 %) contents, and its surface was
covered with vegetation comprising of *Usnea antarctica*, *Deschampsia antarctica*, *Colobanthus quitensis*,
*Ochrolechia frigida*, and different mosses. The older and well developed soil, KGI D, showed distinct contents
of nitrogen (0 – 3 cm: 0.39 %; 3 – 15 cm: 0.03 %), and of soil organic carbon (0 – 3 cm: 3.22 %; 3 – 15 cm: 0.24
%). The complete surface of KGI D was covered with *Deschampsia antarctica*, *Polytrichum spec.*, *Colobanthus*
*quitensis,* and *Usnea antarctica*. The mainly volcanic substrate was not mirrored in the $Al_o+\frac{1}{2}Fe_o$ value, which is
too low to indicate either andic ($\geq 2$ %) or vitric ($\geq 0.4$ %) properties. The $Fe_o/Fe_d$ and $Fe_t/Fe_d$ ratios, and the CIA
did not show a clear direction of pedogenic or chemical weathering in KGI A, B, and C. In contrast, freshly
formed Fe-(hydr-)oxides were indicated by the $Fe_t/Fe_d$ ratio (12.5 – 12.7) in the upper two horizons of KGI D.
The $Fe_o/Fe_d$ ratio also shows a higher activity of Fe(hydr-)oxide formation in the upper horizons by a decreasing
trend with depth in KGI D. The decreasing CIA with depth (51 – 49.1) designates initial silicate weathering
processes combined with the dissolution of Ca, Na and K bearing minerals.
**3.2 Characterization and quantification of the microbial communities**
High-throughput sequencing resulted in a mean of 432,464 reads per sample, ranging from 10,225 (KGI A 10 –
20 cm II) to 744,522 (KGI D 15 – 27 cm II) reads. Rarefaction analysis revealed a sufficient sequencing depth in
all samples for community analysis. The Shannon index showed a decreasing trend in diversity with depth across
all profiles (Table 2), ranging between 4.5 – 4.2 in KGI A, 4.6 – 3.9 in KGI B, 4.8 – 4.0 in KGI C, and 4.8 – 4.3
in KGI D.





[Table 2]
The microbial communities were dominated by 993 bacterial OTUs, which made up 95.2 % to 100 % of the
observed reads in the investigated soils (Figure 3). Looking at the total reads, a large fraction of OTUs is related
the main phyla Proteobacteria (29.7 %), Actinobacteria (28.2 %), Bacteroidetes (11.0 %), Acidobacteria (8.8 %),
Verrucomicrobia (7 %), and Chloroflexi (7 %). Comparing different profiles and soil depths, certain trends
became visible. With depth, the relative abundances of Gemmatimonadetes and Actinobacteria increased, while
the relative abundance of Bacteroidetes and Verrucomicrobia decreased.
[Figure 3]
Generally, KGI A, B, and C showed higher abundances of Actinobacteria and Bacteroidetes, whereas elevated
relative abundances of Acidobacteria and Verrucomibrobia were observed in KGI D. Four OTUs were
associated with Archaea, which were made up exclusively from Nitrososphaeraceae-related organisms within the
phylum Thaumarchaeota and showed relative abundances between 0 % and 4.7 %. Those Thaumarchaeota
showed their highest abundances in the upper 10 cm of KGI B and C, and comparably lower abundances in
deeper soil layers across all profiles.
Soil microbial communities in the individual profiles tended to be different in the uppermost increment, but
displayed an increasing similarity with depth, as shown by a NMDS (Figure 4). Soil depth as well as pH, $C_{org,}$
and $N_t$ were the best factors to explain the respective microbial community structure of the investigated soils.
[Figure 4]
A cluster analysis of the investigated depth intervals, based on the Bray-Curtis dissimilarity, showed the
distribution of abundant OTUs in the different sampling sites (Figure 5). Samples were mainly clustered by the
depth and to a lesser extent by the respective site. The analysis revealed five different clusters of abundant
OTUs.
[Figure 5]
Cluster 1 included seven OTUs (Chloroflexi KD4-96, Rhizobiales, Bradyrhizobium, Pyrinomonadaceae RB41,
Betaproteobacteriales A21b, Candidatus Udaeobacter and Betaproteobacteriales A21b(2)) and was characteristic
for the shallow soil depth (0 – 15 cm) of the site KGI D. Cluster 2 was comprised of 14 OTUs (Tychonema
CCAP 1459-11B, Flavisolibacter, Nitrososphaeraceae, Candidatus Udaeobacter(2), Actinobacteria MB-A2-108,





Blastocatellaceae JGI 0001001-H03, Gaiella, Chthoniobacteer, Nocardioides, Ferruginibacter, Dokdonella,
Candidatus Udaeobacter(3), Rhizobacter, Limnobacter) and characteristic for the upper depth increments (0 – 10
cm) of the sites KGI B and KGI C. Cluster 3 was comprised of 7 OTUs (Ferruginibacter(2), Sphingomonas,
Nocardioides(2), Blastocatella, Sphingomonas(2), Sphingomonas(3), Chloroflexi KD4-96(2)) and characteristic
for the upper depth increments (0 – 10 cm) of site KGI A. The five OTUs associated within cluster 4
(Chitinophagaceae, Chitinophagaceae(2), Pseudarthrobacter, Candidatus Udaeobacter(4), Polaromonas) occurred
in all samples. Cluster 5 was comprised of 22 different OTUs (Microtrichales, Gemmatimonadaceae, Gaiella(2),
Gaiella(3),      Sphingomonas(4),     Gammaproteobacteria,      Gammaproteobacteria(2),      Oryzihumus,
Acidiferrobacteraceae(2),  Nocardioides(3),  Nitrosococcaceae  wb1-P19,  Nitrosococcaceae  wb1-P19(2),
Xanthomonadaceae, Holophagae Subgroup 7, Gaiellales, Gemmatimonadaceae(2), Gemmatimonadaceae(3),
Massilia, Aeromicrobium, Betaproteobacteriales TRA3-20, Pyrinomonadaceae RB41(2), Nitrosomonadaceae
Ellin6067) and was mainly connected to the lower depth increments (10/15 – 40/80 cm) of all sites.
Bacterial abundances determined by the quantification of the 16S rRNA gene showed similar trends across all
profiles and varied in general between $10^4$ and $10^9$ copies g$^{-1}$ soil (Table 2). KGI A (9.1 x $10^8$ copies g$^{-1}$ soil),
KGI B (1.3 x $10^9$ copies g$^{-1}$ soil), KGI C (1.8 x $10^{10}$ copies g$^{-1}$ soil), and KGI D (7.3 x $10^9$ copies g$^{-1}$ soil) had the
highest abundances in the uppermost soil layer. With depth, a substantial decrease in abundances was observed,
resulting in the lowest abundances in KGI B (2.1 x $10^6$ copies g$^{-1}$ soil), KGI C (3.8 x $10^7$ copies g$^{-1}$ soil), and
KGI D (6.3 x $10^7$ copies g$^{-1}$ soil) in the lowermost soil layer. In KGI A, the lowest abundances with 1.74 x $10^4$
copies g$^{-1}$ soil were found in a depth of 10 – 20 cm, before increasing to 1.1 x $10^7$ in the lowermost soil layer.

**4 Discussion**
Glacier forelands provide an excellent opportunity to investigate initial soil formation and its pedochemical and
biological drivers due to the transition from a glacial to a pedogenic geosystem. Over the last 50 years, the
Ecology Glacier on King George Island has retreated 500 m inland (Braun and Gossmann, 2002), exposing a
large area to initial soil-forming processes, periglacial climate and the colonization of microbial pioneers. Our
findings reveal differences in the soil-forming processes and their interaction with the microbial communities on
decadal timescales compared to centennial to millennial timescales.
The investigated soils were characterized by highly diverse microbial communities dominated by Proteobacteria,
Actinobacteria, Acidobacteria, Bacteroidetes, and Verrucomicrobia, which is in accordance with observations in





other Antarctic habitats (e.g. Ganzert et al., 2011; Yan et al., 2017; Meier et al., 2019). Differences between the
sites with regards to the observed community compositions were found only in near-surface substrates in the
upper 10 cm, while the microbial communities became less diverse and more similar with increasing depth
across all investigated soil profiles. Multiple clusters of co-occurring and abundant operational taxonomic units
(OTUs) for different sites and depths (Clusters 1, 2, 3, and 5) as well as a cluster of ubiquitous OTUs (Cluster 4)
were observed. OTUs are the most used basic diversity units in large-scale characterizations of microbial
communities and show high levels of ecological consistency at a global scale (Schmidt et al., 2014). The most
abundant OTUs in Cluster 4 were metabolically flexible organisms, such as OTUs related to Chitinophagaceae
or *Polaromonas*. Several Chitinophagaceae-related OTUs were present in the investigated soils and especially
abundant in the upper depth increments. Chitinophagaceae have been observed in Antarctic soils before (e.g.
Pershina et al., 2018; Dennis et al., 2019) and were described to play a role in the degradation of chitin and other
soil organic compounds (Chung et al., 2012). The degradation of organic compounds in the course of microbial
respiration by Chitinophagaceae and other heterotrophic microorganisms could affect the soil pH and might
enhance the chemical weathering close to the surface.
Additionally, a *Polaromonas*-associated OTU could be observed in all soils and depths (Cluster 4). These
globally occurring organisms are able to survive in a dormant state (Darcy et al., 2011) and are, due to high
levels of horizontal gene transfer, metabolically versatile (Yagi et al., 2009). These organisms are known to
utilize a wide range of substrates such as $H_2$ (Sizova and Panikov, 2007) or diverse organic compounds provided
for instance by sea spray such as acetate, chloroacetate or octane (Mattes et al., 2008) and could be therefore a
pioneer species in the recently exposed soil substrates in the foreland of the Ecology Glacier. In contrast to this
cluster of frequently occurring OTUs, three different clusters consisted of abundant OTUs were found in the
uppermost depth increments of the bare soil (KGI A, Cluster 3), slightly to moderately vegetated soil (KGI B
and KGI C, Cluster 2), and fully vegetated soil (KGI D, Cluster 1). For instance, one important group within
cluster 1 are OTUs related to Rhizobiales or Bradyrhizobium, which are known to be associated with the
rhizosphere of plants, were particularly abundant in the fully vegetated site KGI D. These organisms are
keyplayers for the fixation of nitrogen in soil ecosystems.
The differences in microbial community composition of the four study sites were also reflected in the microbial
diversity of the near-surface depth increments, which increased slightly with vegetation coverage. Before lichens
or vascular plants appear, abundant and diverse microbial communities are known to colonize recently exposed
substrates (Schmidt et al., 2008; Bajerski and Wagner, 2013). These communities are dominated by



photosynthetic, and heterotrophic $N_2$-fixing bacteria (Strauss et al., 2012), resulting in an initial accumulation of
labile carbon and nitrogen pools and play therefore an important role as pioneers for the further development of
the fresh glacier forefield sediments. However, our dataset did not indicate any obvious prokaryotic organisms
associated with phototrophic carbon fixation, but based on the amount of filtered chloroplast-related sequences,
this process is at least partly facilitated by eukaryotic organisms such as algae. Those microbial pioneers
contribute to the stabilization and physical and chemical development of recently exposed substrates (Dietrich
and Perron, 2006; Schulz et al., 2013), and initiate a cascade of crucial processes (e.g. carbon and nitrogen
accumulation or bioweathering) that result in the formation of soils in which complex vegetation can grow
(Ciccazzo et al., 2016).
As mentioned above, microbial pioneer communities play an important role in initial soil formation. They alter
the original soil environment; and are, in turn, influenced by ongoing pedogenic processes, succession, and plant
colonization (Schulz et al., 2013). The site-specific microbial communities and the occurrence of defined clusters
in the upper part of the soil profiles changed according to the vegetation coverage and potentially related soil
properties such as the SOC or the soil pH.
Vegetation can influence the surrounding soil and its properties as well as the present soil microbiome in
different ways, e.g. by releasing plant root exudates (Badri et al., 2009; Chaparro et al., 2013), by plant litter
input (Boy et al., 2016) or by altering thermal and moisture retention of the soil (Almeida et al., 2014). To what
extent the microbial communities in Antarctic soils are directly influenced by vegetation and vice versa is
debated controversially (Yergeau et al., 2007b; Kielak et al., 2008; Teixeira et al., 2010). As vegetation coverage
increased, microbial communities shifted towards plant-related microorganisms in the foreland of the Damma
Glacier in the Alps (Rime et al., 2015). However, we could not observe similar effects on the microbial
communities in the lower part of the investigated soils in the foreland of the Ecology Glacier. This is probably
due to the lack of deeper roots of pioneer plants and the short time since plants colonized the foreland. The effect
of plants on microbial community composition seemed to be limited to the upper part of the soils in the foreland
of the Ecology Glacier since communities in depths > 10 cm were similar in all soil profiles regardless of plant
coverage. Since more developed soils in the ice-free areas of Antarctica did not show mycorrhization, Boy et al.
(2016) concluded that plants influence the colonized soil more by litter input than by direct transfer of
photoassimilates to the surrounding soil. The input of plant litter leads to an increase of soil nitrogen and SOC
contents especially in the upper centimeters of the soil. The succeeding decomposition of organic compounds in
the course of microbial respiration could lead to a decrease of the pH value in the soil. In the upper and even in





the lower part of the investigated soils, soil pH was altered by plant coverage, but shows only little influence on
microbial community structure. In soil environments, pH usually is a significant attribute that shapes the present
microbial community in favor of certain bacterial phyla (Smith et al., 2010; Bajerski and Wagner, 2013; Ganzert
et al., 2014; Siciliano et al., 2014). Our results show that in the foreland of the Ecology Glacier, other vegetation-
related properties, such as the SOC content, or soil moisture, and thermal retention, influence the microbial
communities more significantly close to the surface than the pH value of the soils.
The SOC content has been shown to have a significant influence on microbial communities in cold habitats (e.g.
Bajerski and Wagner, 2013; Ganzert et al., 2014; Rime et al., 2015; Wojcik et al., 2018). After the initial
accumulation of labile carbon and nitrogen pools by microorganisms and the subsequent colonization of plants,
the input of additional soil organic matter in the form of litter might sustain a richer heterotrophic community in
the otherwise nutrient-poor environments of Antarctica. The presence of vegetation has been suggested to
enhance the soil moisture and thermal retention of soils, thus reducing the severity of Antarctic conditions on the
soil environment (Almeida et al., 2014). The soil moisture affects enzymatic and microbial activity (Brockett et
al., 2012), the primary production (McKnight et al., 1999), and ultimately influences the microbial community
structure in a variety of Antarctic habitats (Smith et al., 2010; Niederberger et al., 2015). Another study showed
that soil temperature affects microbial community composition and soil respiration (Yergeau et al., 2012). In
addition to the above-mentioned effects of plant colonization on SOC or soil pH, slightly higher and more stable
moisture and temperature regimes due to the vegetation-related retention could lead to the differences in
community compositions observed in the foreland of the Ecology Glacier, such as increased abundances of
Verrucomicrobia-related species.
The present microbiome was influenced by the soil properties of the upper centimeters, such as the initial
accumulation of SOC and nitrogen and the ongoing soil formation with its initial weathering processes and plant
colonization. Conversely, the microbiome in deeper parts of the soil was affected by a variety of soil chemical
parameters that change with depth (e.g. increase in soil pH, no quantifiable amounts of C and N), which
explained a significant fraction of changes in the composition of the microbial community in the investigated
soils and resulted in different, less abundant, and less diverse microbial communities. Eilers et al. (2012)
compared several soil profiles in a forested montane watershed, where the most variable communities were
located down to a depth of 10 cm and where less diverse and more similar microbial communities could be
observed at depths > 10 cm regardless of the landscape position. They suggested that changes in soil properties
with depth (e.g., pH, organic carbon quantity and quality, differences in temperature or moisture regimes)



represent an ecological filter which makes it difficult for adapted surface-dwelling microorganisms to thrive, and
causes a shift in the community composition in deeper soil horizons. Furthermore, changes in soil
microstructure, induced e.g. by frequent freeze-thaw cycles and associated changes in pore spacing and nutrient
contents have been related to shifts in microbial community compositions in soils from maritime Antarctica
(Meier et al., 2019). Meier et al. (2019) observed a change towards a lenticular microstructure below 20 cm
depth, which was related to significant changes in the microbial community compositions. Some of the observed
OTUs in deeper soils were *Acidiferrobacteraceae*-related organisms, which usually are associated with
autotrophic lifestyles such as sulfur and iron oxidation, and have a broad range of possible substrates, such as
ferrous iron, thiosulfate or ferric iron (Hallberg et al., 2011). In initial soils on James Ross Island, Meier et al.
(2019) found similar OTUs in the lower depth increments and connected those to mineral weathering in the
course of microbial iron cycling. Cryoturbation, a process that would mix topsoil material with deeper soil
horizons and vice versa, was reported to be influential for both abundance and diversity of bacterial communities
in the foreland of the Ecology Glacier (Zdanowski et al., 2013). However, our results indicate that cryoturbation
in these soils is of minor importance since in all soil horizons and at all study sites a clear differentiation with
regard to the community structure with depth was evident.
Depth and soil properties influenced not only microbial diversity and community composition but also microbial
abundances, which increased with vegetation cover and decreased significantly with soil depth. Exponentially
decreasing microbial abundances and biomass with depth are a common observation in soil environments
(Blume et al., 2002a; Eilers et al., 2012). Although the investigated areas and soils are ice-fee for just a few
decades, the bacterial abundances were high ($10^3$ - $10^{10}$ copies g$^{-1}$ soil) showing similar trends across all
investigated soils. Grzesiak et al. (2009) reported > $10^{10}$ counts per gram soil for the foreland of the Ecology
Glacier. These high bacterial abundances are comparable to abundances observed in other parts of the Antarctic
Peninsula (e.g. Meier et al., 2019). A positive relationship between microbial abundances and vegetation as well
as vegetation-related environmental factors (e.g. water content, organic carbon, and nitrogen content) was also
observed by Yergeau et al. (2007b). With increasing soil development along glacier forelands, defined by
increasing carbon and nitrogen contents, decreasing pH, increasing vegetation coverage and increasing
weathering ratios, we observed increasing microbial abundances which is consistent with other observations in
cold environments (e.g. Bajerski and Wagner, 2013; Wojcik et al., 2018). Nevertheless, the relatively high
abundances in the upper centimeters could also be influenced by algae and lichens such as *Usnea antarctica* and
its chloroplasts.



The results show that on a decadal timescale after deglaciation, changes in microbial abundances, community
compositions, and plant coverage are accompanied by lowering of the soil pH, and initial accumulation of SOC
and nitrogen, which are the main soil-forming processes in the soils in the foreland of the Ecology Glacier. In
contrast to these rather rapidly changing parameters, the quantifiable formation of pedogenic oxides and the
increase in chemical weathering require much more time under the current climatic conditions of King George
Island. The initial chemical weathering processes only became evident in the Cambic Cryosol of KGI D, which
is exposed for over 100 years. The main indication is the formation of Fe-(hydr-)oxides and a slight increase of
the CIA at KGI D. On the other hand, the weathering related indices ($Fe_t/Fe_d$ and CIA) did not show a clear
depth differentiation of pedogenic or weathering processes in the recently exposed soils (KGI A, KGI B, KGI
C). Therefore, the chemical properties of the parent material remain almost unaltered.
Generally considered, weathering efficiency is strongly dependent on the ambient temperature (Štyriaková et al.,
2012). Compared to temperate ecosystems, soils in high latitudes form over longer periods of time (Ellis and
Mellor, 1995). Despite the low metabolic activity, soil organisms such as bacteria, fungi, and nematodes promote
soil-forming processes in maritime Antarctica (Bölter, 2011) by driving the nitrogen and carbon cycle (Yergeau
et al., 2007a; Cowan et al., 2011; Barrett et al., 2008), and affecting weathering processes in Antarctic soils (Jie
and Blume, 2002). The biological weathering of rock material is a crucial process that maintains a continuous
supply of inorganic nutrients for prokaryotic and eukaryotic life in barren environments (Adams et al., 1992;
Illmer et al., 1995) and might be of major importance for the ongoing ecological succession towards more
complex communities in recently exposed substrates. Certain prokaryotic genera present in the investigated soils,
such as *Polaromonas* or *Massilia*, were associated with mineral weathering in the past (Caporaso et al., 2010;
Qi-Wang et al., 2011). Frey et al. (2010) showed that such microorganisms could enhance elemental release
from granite by colonizing rock surfaces and lowering the ambient pH by secreting organic acids and hydrogen
cyanide for instance. This process may be also responsible for the lowering of the pH values in the upper two
depth increments of the bare soil KGI A. Subsequently, the respiration of organic matter originating from plant
litter by an active, diverse and abundant heterotrophic community including for example Chitinophagaceae could
further decrease the soil pH, and thus impact weathering rates especially over longer timescales

**5 Conclusions**



This study contributes to a better understanding of the interrelation between microbial communities and soil-
forming processes in recently deglaciated Antarctic soil substrates and the timescales required for such
processes. We found highly diverse communities of microbial pioneers and plants, particularly in the upper part
of soils, formed in the same substrate in the recently (< 50 years), deglaciated foreland of the Ecology Glacier
and behind its lateral moraine (deglaciated >100 years). In the upper depth increments, differences in the soil
chemical and microbiological properties were found even between the three sites in the foreland (KGI A, B, C),
which became ice-free at the same time. Soil pH and SOC depended on the vegetation coverage of the respective
site and especially the soil pH in the vegetated sites could be impacted by microbial degradation of plant litter.
The lowering of the soil pH in the bare soil, however, may be explained by more active Chitinophagaceae and
other potential heterotrophs, and the degradation of organic material of microbial origin, such as chitin from
fungi.
Soil depth represents a variety of changes in the environment such as the increase in soil pH or the decrease in
organic carbon contents and was the strongest determining factor explaining the decrease in microbial diversity
and abundances. The microbial communities were similar at all sites in > 10 cm, regardless of their exposure age
after deglaciation. This means that cryoturbation processes may not have played a major role so far, otherwise
we would not have obtained clear depth functions of soil properties such as SOC and $N_t$ content, or additionally
of the $Fe_d/Fe_t$ ratio and the CIA at the oldest site KGI D.
On a decadal timescale after deglaciation, changes in soil pH, and initial accumulation of soil carbon and
nitrogen were the main soil-forming processes, which were accompanied by changes in microbial abundances,
community compositions, and plant cover. On a centennial to a millennial timescale after deglaciation,
quantifiable silicate weathering and formation of pedogenic (hydr-)oxides could be observed. The cold climate
of Antarctica slows down microbial weathering processes and soil formation rates on recently exposed
sediments. However, we conclude that microbial metabolism is responsible for measurable changes of soil
properties such as pH at a very early stage (within decades) before the soil surface is colonized by pioneer plants
or soil horizons other than C horizons are detectable.

**Data availability.** Demultiplexed raw sequencing data were submitted to the European Nucleotide Archive
(http://www.ebi.ac.uk/ena, last access: 2 June 2020) under accession number PRJEB37594.



**Author Contributions.** PK and DNS designed and conducted fieldwork. PKr and PK contributed to lab data. PKr, PK, and DW wrote the main manuscript. PKr and PK prepared figures. All authors contributed to the interpretation of the results and valuable discussion.

**Competing interests.** The authors declare no competing financial interests.

**Acknowledgments.** The present work was possible with the financial support of the National Council of Technological and Scientific Development (CNPq), Brazil. We thank the Brazilian Navy, TERRANTAR led by Prof. Carlos E.G. Schaefer for all logistics and the colleagues (38. Polska Wyprawa Antarktyczna) from the Henryk Arctowski Polish Antarctic Station for additional support to realize the successful fieldwork during the Antarctic expedition in summer of 2013/2014. We thank Heiner Taubald (University of Tübingen) for XRF measurements. The 16S rRNA gene amplicon (HiSeq) sequencing was financed through the Helmholtz Research Program "Geosystem – The Changing Earth" (GFZ German Research Center for Geosciences, Helmholtz Center Potsdam). We thank Dr. Fabian Horn and Dr. Sizhong Yang for the procession of raw sequencing data (both GFZ). This study was further supported by the Deutsche Forschungsgemeinschaft (DFG) in the framework of the priority program "Antarctic Research with Comparative Investigations in Arctic Ice Areas" by a grant to D.W. (WA 1554/18) and P.K. (KU 1946/8-1).



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





**Table 1: Major soil physical and soil chemical data, CIA and vegetation cover**

| Soil | Depth | Vegetation | BD[a] | Sand | Silt | Clay | $N_t$ | SOC | pH | | Pedogenic Ratios | | CIA[b] |
|---|---|---|---|---|---|---|---|---|---|---|---|---|---|
| | | | | | | | | | | | $Fe_o/$ | $Fe_t/$ | |
| | [cm] | [surface cover in %] | [g cm$^{-3}$] | | [%] | | [%] | [%] | $H_2O$ | $CaCl_2$ | $Fe_d$ | $Fe_d$ | $Al_o*0.5 *Fe_o$ |
| *KGI A S 6°09´991´´,W 58°28´007´´, 38 m a.s.l.* | | | | | | | | | | | | | |
| Hyperskeletic Cryosol | 0-1 | bare soil | n.d. | n.d. | n.d. | n.d. | < 0.03 | < 0.10 | 7.9 | 6.8 | 0.47 | 11.8 | 0.03 | 53.2 |
| | 0-10 | | 1.08 | 50 | 32 | 18 | < 0.03 | < 0.10 | 8.3 | 7.5 | 0.77 | 12.5 | 0.05 | 53.7 |
| | 10-20 | | 1.07 | 55 | 29 | 17 | < 0.03 | < 0.10 | 8.7 | 7.5 | 0.71 | 12.7 | 0.04 | 53.0 |
| | 20-40 | | n.d. | 52 | 28 | 19 | < 0.03 | < 0.10 | 8.9 | 7.8 | 0.54 | 10.2 | 0.06 | 51.2 |
| *KGI B, S 6°09´953´´,W 58°27´852´´, 31 m a.s.l.* | | | | | | | | | | | | | |
| Hyperskeletic Cryosol | 0-1 | *Usnea ant. (90),* | 0.98 | n.d. | n.d. | n.d. | < 0.03 | < 0.10 | 7.4 | 6.6 | 0.21 | 12.6 | 0.02 | 49.9 |
| | 0-10 | *Deschampsia ant. (5),* | 1.07 | 54 | 29 | 16 | < 0.03 | < 0.10 | 7.7 | 6.5 | 0.25 | 10.9 | 0.03 | 50.4 |
| | 10-20 | *Colobanthus quit. (5);* | n.d. | 62 | 23 | 15 | < 0.03 | < 0.10 | 8.3 | 7.2 | 0.19 | 12.2 | 0.02 | 49.9 |
| | 20-80 | *Total coverage 5* | 1.01 | 60 | 25 | 16 | < 0.03 | < 0.10 | 8.5 | 7.4 | 0.19 | 11.3 | 0.02 | 49.5 |
| *KGI C, S 6°09´947´´,W 58°27´862´´,40 m a.s.l.* | | | | | | | | | | | | | |
| Hyperskeletic Cryosol | 0-1 | *Usnea ant. (70),* | n.d. | n.d. | n.d. | n.d. | 0.09 | 1.24 | 6.2 | 5.4 | 0.34 | 11.4 | 0.03 | 50.3 |
| | 0-10 | *Deschampsia ant. (10),* | n.d. | 58 | 28 | 14 | < 0.03 | 0.15 | 7.2 | 6.3 | 0.23 | 11.5 | 0.03 | 50.5 |
| | 10-20 | *Colobanthus quit. (10),* | n.d. | 60 | 27 | 14 | < 0.03 | < 0.10 | 8.1 | 7.0 | 0.31 | 10.9 | 0.04 | 50.2 |
| | 20-40 | *Ochrolechia frigida (5), Mosses (5); Total coverage 80* | n.d. | 59 | 26 | 15 | < 0.03 | < 0.10 | 8.1 | 7.0 | 0.29 | 13.4 | 0.03 | 50.0 |
| *KGI D, S 6°09´976´´,W 58°28´260´´,54 m a.s.l.* | | | | | | | | | | | | | |
| Cambic Cryosol | 0-3 | *Deschampsia ant. (50),* | 0.81 | n.d. | n.d. | n.d. | 0.39 | 3.22 | 5.2 | 4.8 | 0.26 | 12.7 | 0.03 | 51.0 |
| | 3-15 | *Polytrichum spec. (40),* | 0.97 | 62 | 31 | 8 | 0.03 | 0.24 | 6.3 | 5.1 | 0.16 | 12.5 | 0.02 | 50.9 |
| | 15-27 | *Colobanthus quit. (5), Usnea ant. (5); Total* | 0.98 | 62 | 27 | 11 | < 0.03 | < 0.10 | 7.3 | 5.9 | 0.16 | 14.7 | 0.02 | 49.3 |
| | 27-60 | *coverage 100* | 1.10 | 65 | 23 | 12 | < 0.03 | < 0.10 | 7.6 | 6.3 | 0.15 | 14.9 | 0.02 | 49.1 |

[a]) Corrected by coarse material > 2 mm





**Table 2: Bacterial abundances and microbial diversity in four different soil profiles close to the Ecology Glacier, King**
**George Island.**

| sample | Bacterial 16 rRNA copies [gene copies $g^{-1}$ soil] | Shannon's H | Evenness |
|---|---|---|---|
| KGI A 0 - 1 | $9.11 \times 10^8 \pm 6.82 \times 10^8$ | $4.57 \pm 0.15$ | $0.56 \pm 0.04$ |
| KGI A 1 - 10 | $1.78 \times 10^7 \pm 5.95 \times 10^6$ | $4.46 \pm 0.08$ | $0.50 \pm 0.01$ |
| KGI A 10 - 20 | $1.74 \times 10^4 \pm 7.57 \times 10^3$ | $4.46 \pm 0.04$ | $0.55 \pm 0.02$ |
| KGI A 20 - 40 | $1.10 \times 10^7 \pm 1.00 \times 10^6$ | $4.22 \pm 0.03$ | $0.47 \pm 0.00$ |
| KGI B 0 - 1 | $1.29 \times 10^9 \pm 3.06 \times 10^8$ | $4.62 \pm 0.13$ | $0.57 \pm 0.04$ |
| KGI B 1 - 10 | $5.33 \times 10^8 \pm 3.92 \times 10^7$ | $4.63 \pm 0.13$ | $0.60 \pm 0.04$ |
| KGI B 10 - 20 | $1.51 \times 10^7 \pm 3.32 \times 10^6$ | $4.18 \pm 0.37$ | $0.44 \pm 0.09$ |
| KGI B 20 - 80 | $2.06 \times 10^6 \pm 3.00 \times 10^5$ | $3.94 \pm 0.24$ | $0.40 \pm 0.06$ |
| KGI C 0 - 1 | $1.78 \times 10^{10} \pm 1.76 \times 10^9$ | $4.80 \pm 0.08$ | $0.64 \pm 0.03$ |
| KGI C 1 - 10 | $2.20 \times 10^9 \pm 1.08 \times 10^8$ | $4.58 \pm 0.04$ | $0.59 \pm 0.02$ |
| KGI C 10 - 20 | $1.61 \times 10^8 \pm 1.53 \times 10^7$ | $4.02 \pm 0.34$ | $0.37 \pm 0.09$ |
| KGI C 20 - 40 | $3.78 \times 10^7 \pm 2.39 \times 10^6$ | $4.09 \pm 0.07$ | $0.40 \pm 0.01$ |
| KGI D 0 - 3 | $7.27 \times 10^9 \pm 1.24 \times 10^9$ | $4.81 \pm 0.15$ | $0.66 \pm 0.02$ |
| KGI D 3 - 15 | $3.33 \times 10^8 \pm 4.90 \times 10^7$ | $4.13 \pm 0.28$ | $0.40 \pm 0.05$ |
| KGI D 15 - 27 | $1.35 \times 10^8 \pm 1.62 \times 10^7$ | $4.23 \pm 0.15$ | $0.44 \pm 0.04$ |
| KGI D 27 - 60 | $6.30 \times 10^7 \pm 1.50 \times 10^7$ | $4.33 \pm 0.07$ | $0.52 \pm 0.03$ |



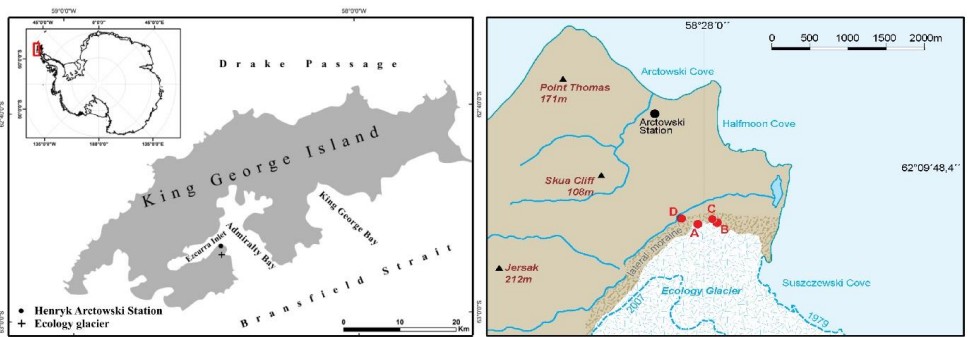


**Figure 1: Location of the study sites. Soil profile locations close to the Ecology Glacier are marked as red dots. Soil**

**profiles A, B, and C are located in the glacier foreland deglaciated since 1956 (Birkenmajer, 2002). Profile D is close to**

**the outer side of the lateral moraine. The dashed blue lines indicate the glacier front in 1979 and 2007 (Source:**

**Orthophotomap from 2007, Department of Antarctic Biology, Polish Academy of Sciences; see also Pudelko, 2008).**






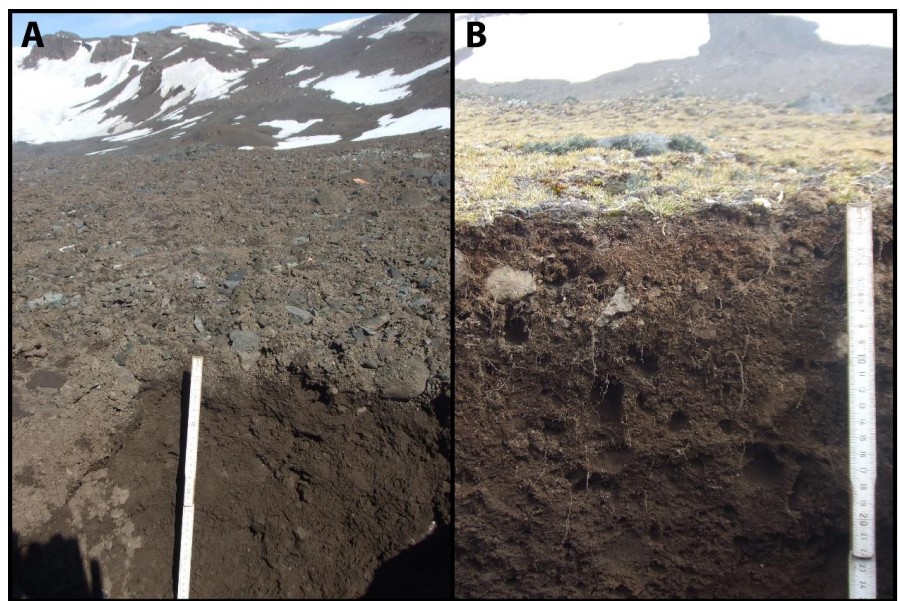


**Figure 2: Photographs of the investigated Cryosols on King George Island, South Shetland Islands. (A) KGI A, a**
**hyperskeletic Cryosol, was located in the foreland of the Ecology Glacier, which was deglaciated for approx. 50 years.**
**(B) Soil profile KGI D, a Cambic Cryosol, was located directly beyond the lateral moraine of the Ecology Glacier and**
**was deglaciated for over 100 years.**




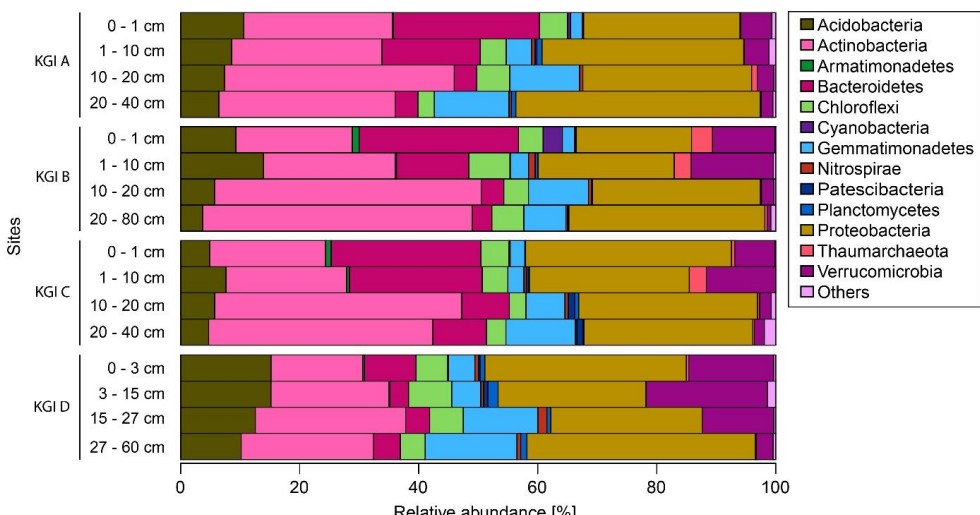

**Figure 3: Relative abundances of phyla of three soil profiles (KGI A, KGI B, KGI C) in the recently deglaciated foreland of the Ecology glacier and one soil profile from behind the lateral moraine (KGI D) on King George Island, South Shetland Islands. Sample triplicates are merged. Only phyla with an abundance of at least 1 % at a given site are presented. Less abundant phyla are summarized as "Others".**





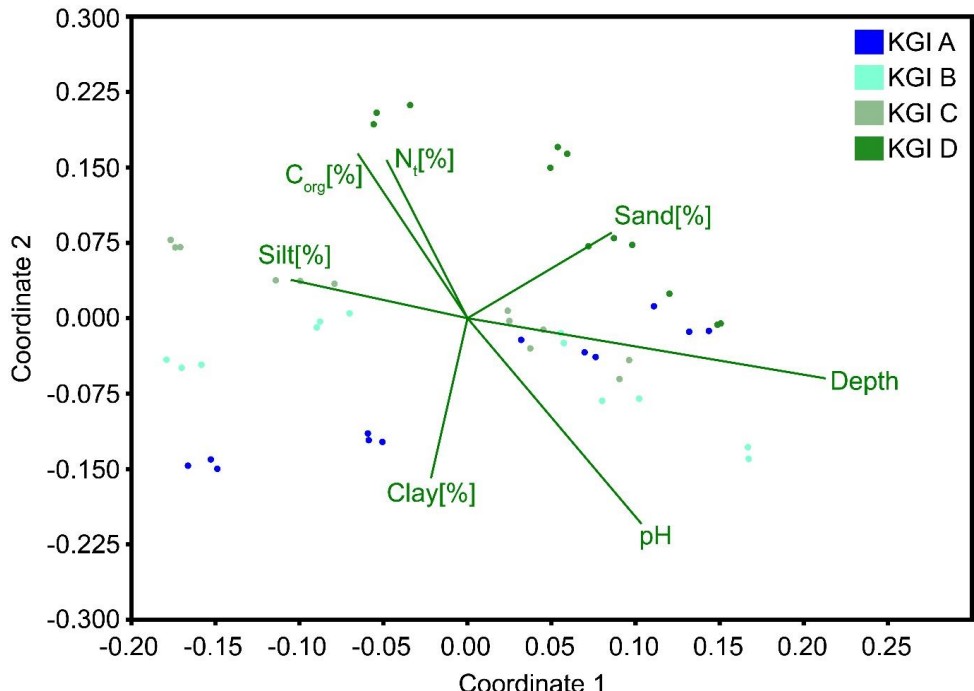


**Figure 4: Non-metric multidimensional scaling plot comparing the microbial communities of three soil profiles in the**

**foreland and one soil profile behind a lateral moraine of the Ecology Glacier, King George Island, based on the Bray-**
**Curtis dissimilarity. Environmental parameters were standardized using z-scores. The stress value was 0.11.**



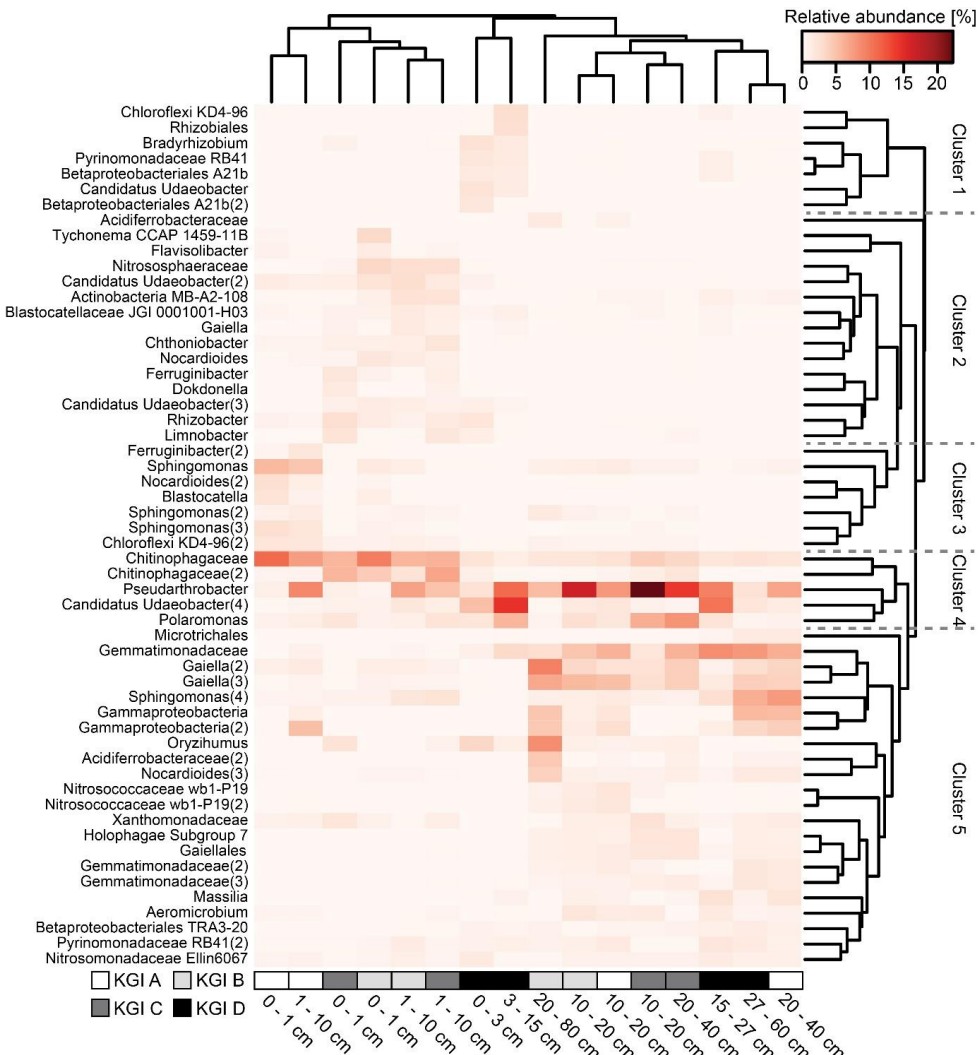


**Figure 5: Heatmap based on the relative abundances of the observed operational taxonomic units (OTUs) in three soil**

**profiles (KGI A, KGI B, KGI C) in the recently deglaciated foreland of the Ecology glacier and one soil profile from**

**behind the lateral moraine (KGI D) on King George Island, South Shetland Islands. Only OTUs with a relative**

**abundance of at least 1.5 % in a given sample are shown. Presented OTUs were clustered using average linkage**

**hierarchical clustering. Samples were clustered based on the whole community using average linkage hierarchical**

**clustering.**