# Peer review of "Influence of microorganisms on initial soil formation along a"

_Biogeosciences, 2020_

## Referee Comment (RC1) · Anonymous Referee #1 · 25 Aug 2020

The manuscript by Krauze et al. examined microbial communities found within recently deglaciated cryosols in Antarctica in order to couple dominant pedogenic processes with microbial community structure. This is an exciting topic within the scope of Biogeosciences and the authors have identified an interesting model system for their study. The manuscript was well written with clear language and was presented nicely as well. Unfortunately, serious flaws within the experimental design and methodology employed for assessing microbial communities have effectively prevented the authors from being able to draw any meaningful, scientifically robust conclusions regarding the microbiota within their system.

[Figure]

The first major problem is the experimental design. There was no experimental replication at any site and because the entire study consisted of four cores and four depths were examined for each core, there were only 16 samples in total. This would not be a problem in a simple system, but this is a complex system with many variables to account for. As soon as a simple factor is taken into account, depth for instance, the effective sample size decreases. Comparisons involving only the top layers, bottom layers or single cores are restricted to four observations. This number is important because regression typically requires five or more observations in order to be able to calculate significance values and a statistical evaluation of regression would allow statements such as: With depth, pH increased (paraphrased from lines 177-178), "The Shannon index showed a decreasing trend in diversity with depth" (line 197), "With depth, the relative abundances of Gemmatimonadetes and Actinobacteria increased, while the relative abundance of Bacteroidetes and Verrucomicrobia decreased" (lines 205-206), "the microbial communities became less diverse and more similar with increasing depth across all investigated soil profiles" (lines 258-259). Without statistics backing up these statements, they constitute opinions, not evidence. I will concede that it is typical to report a handful of opinions and observations, particularly when there is a visual indication of a relationship despite the lack of statistical support. However a close examination of this manuscript reveals that apparently no statements were evaluated using any sort of statistical test. This is not typical or acceptable. As a reader of this manuscript, I don't know what has been evaluated statistically and therefore I don't know what constitutes a scientific statement on the microbial community.

The second problem I identified involved specific methodological choices in the microbial community processing / analysis. I find it difficult to accept that 100% of the microbial community in any cryosol could be classified. This is likely a direct result of the OTU picking method, which involved mapping sequence data to a database. Results like this indicate that the entire community is not being reported. Instead the current analysis compares proportions of classified data, which may or may not represent the community accurately. To know for sure, it is necessary to know how much

of the microbial community is "missing" in the analysis because it did not map to the Silva database. It is often helpful to track the fate of raw data, which requires some additional accounting. The authors should list each dataset in a supplemental table. This table should include, for each individual dataset: SRA object ID, sample ID, number of reads obtained prior to any QC (raw data), number of reads that successfully merged, number of reads that remained after trimmomatic, reads that remained after chimera removal and finally, reads that were classified into OTUs which is the final number of reads that were used for analysis. Negative PCR controls should always be included in such a table.

Actually, I was surprised and somewhat concerned that there was no mention of negative controls. The method of Meier, 2019 used a total of 30 PCR cycles, which is quite high but understandable if the DNA yields are low, which apparently was the case for some samples (lines 151-152). Given the combination of high number of PCR cycles and low biomass, negative controls are absolutely necessary. Table 2 indicates that the deeper soils have much less template for PCR and the authors also concluded that the deeper soils are more similar to one another (line 259). If the deeper samples resemble negative controls due to having a small amount of template, it should come as no surprise that they resemble one another. This very real possibility needs to be ruled out. In addition, it would be quite helpful to include information on DNA extracted per gram of soil for each sample (and replicate) and which samples were extracted three times. The methods just say "samples with low yields". As a side note, I don't understand the point of pooling, since the method of Meier 2019 used a fixed volume of genomic DNA.

Finally, the microbial resesarch community is largely moving beyond OTU-based analysis in favor of amplicon sequence variant (ASV) analysis. The authors may consider switching over to this high-resolution approach that in some cases can provide deep insights into microbial community structure: (https://www.bioconductor.org/packages/devel/bioc/vignettes/dada2/inst/doc/dada2-

intro.html).

---

## Referee Comment (RC2) · Anonymous Referee #2 · 21 Sep 2020

Krauze at al. investigated the interplay between the microbial community and soil formation after glacier retreat at Maritime Antarctica (here: King George Island). This is a fascinating topic which fits very well to the scope of Biogeosciences. The paper is well written and the quality of English language is high (only spelling error I found is in line 364, ice fee instead of ice free). All in all the manuscript is a good and inspiring read. Unfortunately some doubts on the experimental design and the drawn conclusions of the study arise while reading and cloud the pleasure remarkably. The absence of hypotheses is puzzling, especially since the chosen ecological model of very young soils, only a few decades after glacier retreat, should have led to several hypotheses around the topic of temporal dynamics of C and N accumulation and the involved microbial

functional traits, thresholds or tipping points in the system regarding soil development etc. A study only aspiring an objective of "identifying processes" but not having any assumptions on which processes it should look, is probably not meeting contemporary standards of good scientific communication anymore. Further, the experimental design is challenged by a very low repetition number (only one field repetition per age class). In Antarctica, harsh conditions melt down iniotially planned sampling schemes for sure, but only 4 soils in single replication in close vicinity to a research station, at least raises the question, why there was not a valid experimental design possible. The authors additionally should make the age determination of their sites much more transparent as well as the data of vegetation on the sites should be reported in full detail. For instance, the authors discuss the role of vegetation for the microbial community in their manuscript, but write simply "mosses, 5% coverage" in the table. This is an almost useless information, as some Antarctic moss species are nourished by aeolian input without interfering much with the soil, while other moss species deeply penetrate into soil with their rhizoids. Additionally, it is highly unlikely, that only one species of lichens occurs, which suggests erratic identification. The age dating of the sites is the crucial issue for the study. The appearance of both higher plants native to Maritime Antarctica already on the second youngest soils is directly leading to the question if the dating was done correctly, since this appearance is either limited to highly developed soils or ornithogenic soils. As the soil parameters suggest a young soil, not having enough N for maintaining populations of higher plants on the long run, it is very likely, that these encounters are due to the frequently seen bird dropping effect (excrements containing seed of Deschampsia or Colobanthus stage the appearance of these plants for a season or two, until the N from the excrement is used up). A coverage of 10% of higher plants on such young soils suggest a high frequency of bird visits, which was excluded by the authors, as this would largely question the results of a real soil development situation of the microbial community, as higher plants alien to the respective community would have been introduced. With the given data all this stays mere speculation, of course. To solve this issue, at least a much better map of the sites, ideally

a satellite map showing the features of the sites as well as their surroundings, along with a concise description of the found vegetation should be offered by the authors, please. If possible some proof of the assumed age, for instance by a combination by 14C and 15N analyses of the soils, showing the age of organic carbon and the source of N (fixed from air or carried in by birds?) would help explain this very rare combination of observed features reported here. These aspects should be discussed in detail, too. Regarding the DNA-based identification of bacteria it stays largely unclear to the reader where the usually pretty high number of unidentified OTUs (so to say the bacteria not yet known to science) in soils of Maritime Antarctica has gone. The abundance graph reads as if the authors could attribute every single OTU to a phylum. In the discussion, as already indicated by the missing hypotheses, the reader misses a true discussion of the "identification of processes" promised in the objectives. Not even the "influences of microorganisms on soil formation" promised by the title were discussed in detail, what anyway would have been impossible by a study leaving the fungi completely aside. After what is discussed in the paper, this study is more investigating the influence of vegetation on the bacterial community of young soils in Maritime Antarctica. A big step towards the originally intended process elucidation of soil formation would be for instance a detailed discussion on the functional traits and abundances of bacteria involved in e.g. C and N accumulation or weathering processes. In the current form, the study is a family list of bacteria of four Antarctic soils, trying to mimic process identification by naming rather weak coincidences. There is surely more to it.

---

## Author Comment (AC1) · 23 Nov 2020

We would like to thank Reviewer #1 for her/his time, effort, and valuable comments. We have
prepared a response taking into account all the points raised, as described below. We show
the reviewer's comments in bold, while our responses are formatted as standard text. Line
numbers refer to the original manuscript.

**The manuscript by Krauze et al. examined microbial communities found within recently**
**deglaciated cryosols in Antarctica in order to couple dominant pedogenic processes**
**with microbial community structure. This is an exciting topic within the scope of**
**Biogeosciences and the authors have identified an interesting model system for their**
**study. The manuscript was well written with clear language and was presented nicely**
**as well. Unfortunately, serious flaws within the experimental design and methodology**
**employed for assessing microbial communities have effectively prevented the authors**
**from being able to draw any meaningful, scientifically robust conclusions regarding the**
**microbiota within their system. The first major problem is the experimental design.**
**There was no experimental replication at any site and because the entire study**
**consisted of four cores and four depths were examined for each core, there were only**
**16 samples in total. This would not be a problem in a simple system, but this is a**
**complex system with many variables to account for. As soon as a simple factor is taken**
**into account, depth for instance, the effective sample size decreases. Comparisons**
**involving only the top layers, bottom layers or single cores are restricted to four**
**observations. This number is important because regression typically requires five or**
**more observations in order to be able to calculate significance values and a statistical**
**evaluation of regression would allow statements such as: With depth, pH increased**
**(paraphrased from lines 177-178), "The Shannon index showed a decreasing trend in**
**diversity with depth" (line 197), "With depth, the relative abundances of**
**Gemmatimonadetes and Actinobacteria increased, while the relative abundance of**
**Bacteroidetes and Verrucomicrobia decreased" (lines 205-206), "the microbial**
**communities became less diverse and more similar with increasing depth across all**
**investigated soil profiles" (lines 258-259). Without statistics backing up these**
**statements, they constitute opinions, not evidence. I will concede that it is typical to**
**report a handful of opinions and observations, particularly when there is a visual**
**indication of a relationship despite the lack of statistical support. However a close**
**examination of this manuscript reveals that apparently no statements were evaluated**
**using any sort of statistical test. This is not typical or acceptable. As a reader of this**
**manuscript, I don't know what has been evaluated statistically and therefore I don't**
**know what constitutes a scientific statement on the microbial community.**

We agree that additional biological replicates would increase the significance of our
observations. With technical triplicates of every sample for the microbiological methods, we
tried to cover spatial heterogeneity of the investigated soils, which worked out well (see. Fig.
4). Anyway, to account for the low number of replications any conclusion drawn from our
statistics was formulated in a careful fashion in our manuscript. In order not to be limited by
the quantity of replicates, we did not compare or made statements regarding single sites or
depth increments.

The statements mentioned in your comment are related to changes we observed across the
whole data set (e.g. depth-dependent changes in pH or microbial diversity) taking into account
16 independent samples. In our opinion, this number of replicates allows for the statements
we made. The same should be true for the NMDS, which highlights environmental factors
shaping the microbial communities (Fig. 4).

**The second problem I identified involved specific methodological choices in the microbial community processing / analysis. I find it difficult to accept that 100% of the microbial community in any cryosol could be classified. This is likely a direct result of the OTU picking method, which involved mapping sequence data to a database. Results like this indicate that the entire community is not being reported. Instead the current analysis compares proportions of classified data, which may or may not represent the community accurately. To know for sure, it is necessary to know how much of the microbial community is "missing" in the analysis because it did not map to the Silva database. It is often helpful to track the fate of raw data, which requires some additional accounting. The authors should list each dataset in a supplemental table. This table should include, for each individual dataset: SRA object ID, sample ID, number of reads obtained prior to any QC (raw data), number of reads that successfully merged, number of reads that remained after trimmomatic, reads that remained after chimera removal and finally, reads that were classified into OTUs which is the final number of reads that were used for analysis.**

Thank you for your suggestion. A table with the relevant information in the supplement was added (Tab. S4) and referred to in the beginning of "3.2 Characterization and quantification of the microbial communities" in the Results section.

Originally, completely unassigned OTUs have been identified, but were very low in their abundance and summarized with other low abundant phyla under the term "Others" in Figure 3. Our data was reanalysed using the amplicon sequence variant (ASV) approach (see the last comment). Similar to the OTU analysis, this approach resulted in low abundances of completely unassigned sequences (minimum: 0.01% in KGI A 10 – 20 cm; maximum: 0.28% in KGI A 0 – 1 cm). Low abundances of unassigned reads in Antarctic soils are common in recent literature (0.28 % of the total data set of Meier et al., 2019; 1.1 % of the total data set in Kim et al., 2019). The very low abundance of unassigned sequences does not mean that an identification to genus/species level for the remaining sequences was possible, though. Many of the OTUs/ASVs shown in Fig. 5, which represent the most abundant reads in the data set, were just classified to the order or family level.

**Negative PCR controls should always be included in such a table. Actually, I was surprised and somewhat concerned that there was no mention of negative controls. The method of Meier, 2019 used a total of 30 PCR cycles, which is quite high but understandable if the DNA yields are low, which apparently was the case for some samples (lines 151-152). Given the combination of high number of PCR cycles and low biomass, negative controls are absolutely necessary. Table 2 indicates that the deeper soils have much less template for PCR and the authors also concluded that the deeper soils are more similar to one another (line 259). If the deeper samples resemble negative controls due to having a small amount of template, it should come as no surprise that they resemble one another. This very real possibility needs to be ruled out.**

A negative control was of course analysed, but not mentioned in the manuscript. The extraction included a negative control, which was afterwards handled like a sample: The amount of DNA was measured by using a Qbit; the extract was used as a template in the PCR; and the resulting PCR product sent for sequencing.

Relevant information was added to "2.4 Nucleic acids extraction" in the Material and Methods section:

"In addition, a negative control without any template but the material and chemicals of the extraction kit was included." (L. 152)

We made minor changes to "2.5 Illumina HiSeq-Sequencing" in the Material and Methods section:

"Total genomic extracts of each sample as well as an extraction negative control and a positive control (*Escherichia coli*) were sequenced using tagged 515F (5'-GTGCCAGCMGCCGCGGTAA-3') and 806R (5'-GGACTACHVGGGTWTCTAAT-3') primers after Caporaso et al. (2010)." (L. 155ff)

Additionally, we now not only refer to the sequencing data set in the ENB, but provide an ASV table (Tab. S5) in the supplement, including all analysed replicates, and both positive and negative controls. We refer to this table in "3.2 Characterization and quantification of the microbial communities" in the Results section. We hope the increased transparency is helpful for the evaluation of our sequencing data.

Looking at the data, the samples with very low biomass (and therefore very low DNA concentrations) do not resemble the community found in the negative control. The most abundant sequences in the negative control were classified as Cutibacterium, Corynebacterium, Staphylococcus, Methylophilaceae, Undibacterium, Pseudomonas and Streptococcus, which had relative abundances >5 % in the negative control. None of these taxa made up more >0.6 % of the total community after merging the triplicates of a sample. Some taxa found in the negative control occur as possible contaminants in some samples, but in very low relative abundances.

We included the negative control in a NMDS. It is located quite far from the cluster of analysed samples, including the deeper ones, and therefore differs significantly in its community composition. With this, we are confident that the deeper samples do not reflect a contamination during extraction and the following steps, but represent the actual community.

**In addition, it would be quite helpful to include information on DNA extracted per gram of soil for each sample (and replicate) and which samples were extracted three times. The methods just say "samples with low yields".**

We are happy to include a table (Tab. S3) showing the amount of soil used for DNA extraction for every replicate in the supplementary material. "Samples with low yields" were highlighted. This table is referred to in "2.4 Nucleic acids extraction" in the Material and Methods section.

**As a side note, I don't understand the point of pooling, since the method of Meier 2019 used a fixed volume of genomic DNA.**

As described in Meier et al. (2019), a fixed volume of DNA extract was used in the PCR, and a fixed amount of DNA of every PCR product was used for sequencing. The mention of pooling was solely referred to the pretreatment of the "samples with low yields" before molecular biological work.

Every replicate of those samples was extracted in triplicates, and the resulting extracts of a sample merged afterwards. Vacuum centrifugation reduced the volume of this pool to 50 µl. This step was necessary for the following molecular biological work (PCR, qPCR) to work.

We hope to clarify this by rewording the relevant part in "2.4 Nucleic acid extraction" in the Material and Methods section as follows:

"Sample replicates with very low DNA yields (Tab. S3) were extracted in triplicates. These
extraction triplicates of a sample replicate were merged and after reducing their volume to 50
µl by vacuum centrifugation ready for the following molecular biological work." (L. 151f)

**Finally, the microbial resesarch community is largely moving beyond OTUbased**
**analysis in favor of amplicon sequence variant (ASV) analysis. The authors may**
**consider switching over to this high-resolution approach that in some cases can**
**provide deep insights into microbial community structure:**
**(https://www.bioconductor.org/packages/devel/bioc/vignettes/dada2/inst/doc/dada2-**
**intro.html).**

We agree. As recommended, we switched to the ASV analysis and reanalysed the sequencing
data using this more modern approach. Additionally, the taxonomical classification was done
using the latest release of the SILVA database (138.1)

To account for the new method, we have rewritten 2.6 Bioinfortmatics and statistical analysis
in the Material and Methods section as follows:

"Raw sequencing data obtained by Illumina HiSeq (2 x 300 bp) were checked for quality with
FastQC (Andrews et al., 2010). The data was demultiplexed by using the *make.contigs*
function in Mothur (version 1.39.5; pdiff = 2, bdiff = 1, and default setting for others; Schloss et
al., 2009). According to the resulting report files, a filtering step was implemented to get fastq
sequence identifiers for sequences with a minimum overlap of >25 bases, maximum
mismatches of <5 bases and no ambigious bases. Next, these sequences were extracted with
the *filterbyname.sh* function from BBTools (Bushnell et al., 2014) from the raw paired-end fastq
file. With QIIME1, sequence orientation was checked and corrected by using the
*extract_barcodes.py* function and the primers were removed using the *awk* command
(Caporaso et al., 2010). DADA2 was used for filtering, dereplication, chimera check, sequence
merge, and amplicon sequence variants (ASV) calling (Callahan et al., 2016). The output of
DADA2 was taxonomically classified by using QIIME2 (Bolyen et al., 2019) and USEARCH
(Edgar, 2010) with SILVA138 (Quast et al., 2013). Resulting data were visualized using R and
PAST4 (Hammer et al., 2001)." (L. 160 – 163)

All figures based on the sequencing data as well as the calculation of the diversity indices were
redone based on the new ASV table. Excluding some changes in their names, the relative
abundances of the presented phyla (see Fig. 3) were not affected substantially by switching
over to the ASV analysis.

**Table S3: Amount of soil used for DNA extraction and resulting DNA concentrations.**

| sample | soil used for extraction [mg] | DNA concentration [ng/µl] |
|---|---|---|
| KGI_A_0_1_a | 557 | 7.13 |
| KGI_A_0_1_b | 550 | 8.2 |
| KGI_A_0_1_c | 530 | 6.87 |
| KGI_A_1_10_a | 510 | 0.121 |
| KGI_A_1_10_b | 564 | 0.213 |
| KGI_A_1_10_c | 585 | 0.103 |
| KGI_A_10_20_a | 1560 | n. d. |
| KGI_A_10_20_b | 1585 | n. d. |
| KGI_A_10_20_c | 1631 | n. d. |
| KGI_A_20_40_a | 1510 | 0.4 |
| KGI_A_20_40_b | 1614 | 0.511 |
| KGI_A_20_40_c | 1574 | 0.376 |
| KGI_B_0_1_a | 505 | 13.3 |
| KGI_B_0_1_b | 494 | 10.1 |
| KGI_B_0_1_c | 541 | 9.13 |
| KGI_B_1_10_a | 590 | 4.61 |
| KGI_B_1_10_b | 590 | 4.83 |
| KGI_B_1_10_c | 549 | 4.11 |
| KGI_B_10_20_a | 592 | 0.09 |
| KGI_B_10_20_b | 538 | 0.04 |
| KGI_B_10_20_c | 529 | 0.08 |
| KGI_B_20_80_a | 1679 | 0.07 |
| KGI_B_20_80_b | 1665 | 0.09 |
| KGI_B_20_80_c | 1639 | 0.04 |
| KGI_C_0_1_a | 485 | 78.4 |
| KGI_C_0_1_b | 500 | 69.6 |
| KGI_C_0_1_c | 550 | 61.2 |
| KGI_C_1_10_a | 548 | 18.5 |
| KGI_C_1_10_b | 544 | 16.5 |
| KGI_C_1_10_c | 553 | 12.4 |
| KGI_C_10_20_a | 530 | 0.8 |
| KGI_C_10_20_b | 570 | 1.17 |
| KGI_C_10_20_c | 565 | 1.13 |
| KGI_C_20_40_a | 550 | 0.273 |
| KGI_C_20_40_b | 570 | 0.428 |
| KGI_C_20_40_c | 562 | 0.101 |
| KGI_D_0_3_a | 552 | 49.6 |
| KGI_D_0_3_b | 505 | 49.4 |
| KGI_D_0_3_c | 585 | 22.6 |
| KGI_D_3_15_a | 560 | 5.46 |
| KGI_D_3_15_b | 514 | 3.87 |
| KGI_D_3_15_c | 512 | 4.97 |
| KGI_D_15_27_a | 562 | 1.35 |
| KGI_D_15_27_b | 561 | 1.27 |
| KGI_D_15_27_c | 586 | 1.37 |

| KGI_D_27_60_a | 599 | 0.322 |
| KGI_D_27_60_b | 550 | 0.337 |
| KGI_D_27_60_c | 548 | 0.501 |

**Table S4: Number of sequencing reads after each processing step.**

| sample | input | filtered | denoised | merged | Non-chimera | 0.01% cutoff |
|---|---|---|---|---|---|---|
| | | | number of reads | | | |
| KGI_A_0_1_a | 823555 | 760184 | 744842 | 728869 | 722825 | 706168 |
| KGI_A_0_1_b | 416652 | 386897 | 378089 | 368092 | 365389 | 359124 |
| KGI_A_0_1_c | 923881 | 848435 | 824280 | 799372 | 786254 | 772324 |
| KGI_A_1_10_a | 473836 | 436670 | 427179 | 416676 | 413400 | 410050 |
| KGI_A_1_10_b | 234950 | 218204 | 212241 | 204825 | 203788 | 201502 |
| KGI_A_1_10_c | 352840 | 330275 | 323230 | 316343 | 314148 | 311452 |
| KGI_A_10-20_a | 36923 | 33865 | 29718 | 26130 | 25958 | 25922 |
| KGI_A_10-20_b | 18966 | 17275 | 14209 | 12268 | 12259 | 12252 |
| KGI_A_10-20_c | 25875 | 23729 | 19871 | 17313 | 17308 | 17300 |
| KGI_A_20_40_a | 905510 | 839764 | 828694 | 814373 | 808267 | 790148 |
| KGI_A_20_40_b | 846168 | 790425 | 779458 | 764252 | 758482 | 737591 |
| KGI_A_20_40_c | 893543 | 830439 | 821643 | 808578 | 801514 | 778446 |
| KGI_B_0_1_a | 802491 | 738909 | 726606 | 711994 | 705038 | 683957 |
| KGI_B_0_1_b | 767042 | 707072 | 685728 | 660207 | 650385 | 634947 |
| KGI_B_0_1_c | 776140 | 717207 | 691876 | 662511 | 644483 | 630913 |
| KGI_B_1-10_a | 870220 | 801229 | 775510 | 742168 | 726536 | 700009 |
| KGI_B_1-10_b | 763113 | 708362 | 698110 | 683584 | 679585 | 655095 |
| KGI_B_1-10_c | 984339 | 904070 | 867753 | 821445 | 790478 | 763807 |
| KGI_B_10_20_a | 902645 | 835484 | 829409 | 818332 | 809608 | 796113 |
| KGI_B_10_20_b | 50853 | 47043 | 41987 | 37904 | 37729 | 37613 |
| KGI_B_10_20_c | 45144 | 41598 | 36756 | 32746 | 32618 | 32557 |
| KGI_B_20_80_a | 797077 | 739614 | 734477 | 727607 | 713996 | 706767 |
| KGI_B_20_80_b | 666167 | 617867 | 613646 | 605338 | 594001 | 587915 |
| KGI_B_20_80_c | 59859 | 55236 | 50726 | 47120 | 46730 | 46649 |
| KGI_C_0_1_a | 748549 | 692351 | 673800 | 648601 | 635472 | 605915 |
| KGI_C_0_1_b | 896152 | 828214 | 784842 | 728938 | 687678 | 661043 |
| KGI_C_0_1_c | 780292 | 724237 | 701902 | 672607 | 650189 | 619970 |
| KGI_C_1_10_a | 1072129 | 979738 | 941569 | 893583 | 866371 | 830430 |
| KGI_C_1_10_b | 796453 | 734715 | 714128 | 686413 | 675197 | 651021 |
| KGI_C_1_10_c | 945868 | 879402 | 852158 | 818052 | 799307 | 765149 |
| KGI_C_10_20_a | 914463 | 847130 | 837834 | 821943 | 814309 | 788037 |
| KGI_C_10_20_b | 724047 | 663724 | 657595 | 646501 | 642849 | 625122 |
| KGI_C_10_20_c | 379777 | 348578 | 342785 | 334296 | 331763 | 325740 |
| KGI_C_20_40_a | 805325 | 743983 | 737314 | 727505 | 720592 | 705527 |
| KGI_C_20_40_b | 883724 | 818626 | 808175 | 789997 | 782377 | 762057 |
| KGI_C_20_40_c | 651443 | 603352 | 595920 | 583664 | 578353 | 564023 |
| KGI_D_0_3_a | 779880 | 715854 | 698512 | 675201 | 664504 | 626027 |
| KGI_D_0_3_b | 787246 | 723529 | 685098 | 643359 | 617163 | 586001 |
| KGI_D_0_3_c | 898392 | 826444 | 770393 | 701924 | 643856 | 619137 |

| | | | | | | |
|---|---|---|---|---|---|---|
| KGI_D_3_15_a | 815335 | 754410 | 739659 | 718535 | 709893 | 677501 |
| KGI_D_3_15_b | 950935 | 871121 | 850341 | 820666 | 806893 | 766920 |
| KGI_D_3_15_c | 930601 | 858951 | 806053 | 732072 | 650658 | 627047 |
| KGI_D_15_27_a | 913147 | 851119 | 839448 | 822669 | 812036 | 784157 |
| KGI_D_15_27_b | 1074041 | 989909 | 980950 | 964946 | 953467 | 917368 |
| KGI_D_15_27_c | 869976 | 799507 | 791364 | 777160 | 770119 | 749250 |
| KGI_D_27_60_a | 740686 | 687100 | 681491 | 671687 | 666209 | 645648 |
| KGI_D_27_60_b | 768476 | 714698 | 707202 | 695134 | 689840 | 671215 |
| KGI_D_27_60_c | 852927 | 790621 | 782334 | 769492 | 763600 | 737767 |
| Negative control | 11045 | 9999 | 9955 | 9899 | 9899 | 9819 |
| Positive control | 651120 | 590140 | 589778 | 585268 | 585268 | 585250 |

References

Andrews, S.: FastQC: a quality control tool for high throughput sequence data, 2010.

Bolyen, E., Rideout, J. R., Dillon, M. R., Bokulich, N. A., Abnet, C. C., Al-Ghalith, G. A., ... &
Bai, Y.: Reproducible, interactive, scalable and extensible microbiome data science using
QIIME 2. Nature biotechnology, 37(8), 852-857, 2019.

Bushnell, B.: BBTools software package, 2014.

Callahan, B. J., McMurdie, P. J., Rosen, M. J., Han, A. W., Johnson, A. J. A., & Holmes, S. P.:
DADA2: high-resolution sample inference from Illumina amplicon data. Nature methods, 13(7),
581-583, 2016.

Caporaso, J. G., Kuczynski, J., Stombaugh, J., Bittinger, K., Bushman, F. D., Costello, E. K.,
Fierer, N., Peña, A. G., Goodrich, J. K., Gordon, J. I., Huttley, G. A., Kelley, S. T., Knights, D.,
Koenig, J. E., Ley, R. E., Lozupone, C. A., McDonald, D., Muegge, B. D., Pirrung, M., Reeder,
J., Sevinsky, J. R., Turnbaugh, P. J., Walters, W. A., Widmann, J., Yatsunenko, T., Zaneveld,
J., and Knight, R.: QIIME allows analysis of high-throughput community sequencing data,
Nature methods, 7, 335–336, 10.1038/nmeth.f.303, 2010.

Edgar, R.: Usearch, 2010

Kim, M., Lim, H. S., Hyun, C. U., Cho, A., Noh, H. J., Hong, S. G., and Kim, O. S.: Local-scale
variation of soil bacterial communities in ice-free regions of maritime Antarctica, Soil Biology
and Biochemistry, 133, 165-173, 2019.
Meier, L. A., Krauze, P., Prater, I., Horn, F., Schaefer, C. E. G. R., Scholten, T., Wagner, D.,
Mueller, C. W., and Kühn, P.: Pedogenic and microbial interrelation in initial soils under
semiarid climate on James Ross Island, Antarctic Peninsula region, Biogeosciences, 16,
2481–2499, 10.5194/bg-16-2481-2019, 2019.
Quast, C., Pruesse, E., Yilmaz, P., Gerken, J., Schweer, T., Yarza, P., Peplies, J.,
and Glöckner, F. O.: The SILVA ribosomal RNA gene database project: improved data
processing    and    web-based    tools,    Nucleic    acids    research,    41,    D590-596,
10.1093/nar/gks1219, 2013.
Schloss, P. D., Westcott, S. L., Ryabin, T., Hall, J. R., Hartmann, M., Hollister, E. B.,
Lesniewski, R. A., Oakley, B. B., Parks, D. H., Robinson, C. J., Sahl, J. W., Stres, B., Thallinger,
G. G., Van Horn, D. J. and Weber, C. F.: Introducing mothur: open-source, platform-
independent, community-supported software for describing and comparing microbial
communities. Applied and environmental microbiology, 75(23), 7537-7541, 2009.

---

## Author Comment (AC2) · 23 Nov 2020

We would like to thank Reviewer #2 for her/his time, effort, and valuable comments. We have
prepared a response taking into account all the points raised, as described below. We show
the reviewer's comments in bold, while our responses are formatted as standard text. Line
numbers refer to the original manuscript.

**Krauze at al. investigated the interplay between the microbial community and soil**
**formation after glacier retreat at Maritime Antarctica (here: King George Island). This is**
**a fascinating topic which fits very well to the scope of Biogeosciences. The paper is**
**well written and the quality of English language is high (only spelling error I found is in**
**line 364, ice fee instead of ice free).**

Thank you, we corrected the spelling error.

**All in all the manuscript is a good and inspiring read. Unfortunately some doubts on the**
**experimental design and the drawn conclusions of the study arise while reading and**
**cloud the pleasure remarkably. The absence of hypotheses is puzzling, especially since**
**the chosen ecological model of very young soils, only a few decades after glacier**
**retreat, should have led to several hypotheses around the topic of temporal dynamics**
**of C and N accumulation and the involved microbial functional traits, thresholds or**
**tipping points in the system regarding soil development etc. A study only aspiring an**
**objective of "identifying processes" but not having any assumptions on which**
**processes it should look, is probably not meeting contemporary standards of good**
**scientific communication anymore.**

Thanks for your comment. In order to clarify the objectives of the study, we have reformulated
the corresponding part in the introduction. In addition, we changed certain passages in the text
that refer to "processes".

"We hypothesize that prokaryotic microorganisms initiate/drive soil properties changes (e.g.
soil organic carbon accumulation, weathering) within decades after deglaciation. To test our
hypothesis, we related the investigated soil properties to microbial community structure and
microbial abundances in the recently (< 50 years) deglaciated foreland and compared it with
an older soil located behind a lateral moraine (> 70 years) of the Ecology Glacier." (L. 78ff)

We updated the Conclusions section of our manuscript addressing the hypothesis, and a
potential setup and study design to further study the investigated processes.

"However, we conclude that prokaryotic microorganisms initiate measurable changes of soil
properties such as pH at a very early stage (within decades) before the soil surface is colonized
by pioneer plants or soil horizons other than C horizons are detectable, and thereby promote
weathering processes." (L. 426ff)

"To further verify our conclusions and illuminate the microbial processes driving soil formation,
in future studies multiple comparable setups (freshly deglaciated material vs. older, more
matured soil close to the foreland) could be studied and include metagenomic and
metatranscriptomic analyses." (L. 428)

**Further, the experimental design is challenged by a very low repetition number (only one field repetition per age class). In Antarctica, harsh conditions melt down iniotially planned sampling schemes for sure, but only 4 soils in single replication in close vicinity to a research station, at least raises the question, why there was not a valid experimental design possible.**

In this study, we do not present four age classes represented by one profile each. By sampling three profiles in close proximity in the recently deglaciated area we tried to capture the potential heterogeneity of these soils, and compare these younger soils with an older soil just behind the lateral moraine.

To clarify this, we rephrased a sentence in the Introduction section:

"To capture the heterogeneity of the soil landscape, three soils in close proximity (maximum distance of 150 m) formed on the same substrate and in a similar topographic position but with differing vegetation cover were sampled. These soils, which represent a recently deglaciated area, were compared with an older soil that had formed on a similar substrate that had been deglaciated for at least 70 years." (L. 81ff)

Additionally, we rephrased a sentence in the Material and Methods section:

"The profiles KGI A, B, and C are located within 150 m distance on the same substrate deglaciated since 1979." (L. 117)

In addition, by changing the date we corrected a related mistake in the caption of Fig. 1:

"A, B, and C are located in the glacier foreland deglaciated since 1979."

**The authors additionally should make the age determination of their sites much more transparent as well as the data of vegetation on the sites should be reported in full detail. For instance, the authors discuss the role of vegetation for the microbial community in their manuscript, but write simply "mosses, 5% coverage" in the table. This is an almost useless information, as some Antarctic moss species are nourished by aeolian input without interfering much with the soil, while other moss species deeply penetrate into soil with their rhizoids.**

Many thanks for this comment. Additional $^{14}$C datings were done.
To determine the age of the soil organic carbon, two samples with the highest SOC contents were chosen for AMS $^{14}$C dating (KGI C 0 – 1 cm in the glacier foreland and KGI D 0 - 3 cm from behind the lateral moraine). KGI C 0 – 1 cm was the only sample from the foreland with detectable SOC. The samples were pre-treated with the acid/alkali/acid method and the alkali soluble organics (also known as the humin fraction, i.e. the oldest portion of soil organic matter) and were measured by Beta Analytic, Inc. in Florida, USA.

KGI C (Beta – 570459); IRMS δ$^{13}$C: -27.9 ‰
(92.2%) 1992 - 1995 cal AD (-43 - -46 cal BP)
(3.2%) 1957 cal AD (-8 cal BP)

KGI D (Beta – 570458) IRMS δ$^{13}$C: -26.1 ‰
(95.4%) 1954 - 1956 cal AD (-5 - -7 cal BP)

The results show with a probability of 92.2% that the humin fraction of soil organic matter in profile KGI C was formed after the melting of the glacier in 1979, whereas the humin fraction of the upper 3 cm in profile KGI D (with a probability of 95.4%) is dated 1954 - 1956 cal AD.

The SOC in the lower part of the KGI D may even be older. Therefore, the SOC at KGI D was
already formed at a time when the present foreland of the Ecology Glacier and with this the
sites of KGI A, B, C were still under the glacier.

We included these results in a table in the supplementary (Tab. S1) and integrated the new
information on the potential age of the sites in 3.1 Soil classification and soil properties in the
Results section:

"$^{14}$C dating showed that the humin fraction of soil organic matter in profile KGI C was formed
after the melting of the glacier in 1979 (92.2% 1992 - 1995 cal AD (-43 - -46 cal BP); 3.2%
1957 cal AD (-8 cal BP)), whereas the humin fraction of the upper 3 cm in profile KGI D is
much younger (95.4% 1954 - 1956 cal AD (-5 - -7 cal BP)) (Tab. S1)." (L. 187)

With this first rough listing of plants and the degree of coverage we wanted to test whether
different degrees of coverage are reflected in changes of soil properties. We discussed the
influence of increasing/decreasing vegetation coverage on microbial communities in lines
304ff. Much to our regret, we are not able give a more detailed description of the vegetation,
because we had no botanist in our group. We are aware that a detailed description of the
vegetation would help to better understand the mechanisms of soil processes associated with
specific plants (e.g. different moss species). However, this was not the aim of this study and
we do not discuss this issue. To answer this question would have required a different sampling
approach (1 cm depth increments in the upper 10 cm), which would have been plant species
specific. This can be a next step and should be part of the goals of our next field trip. Again,
many thanks for this comment.

**Additionally, it is highly unlikely, that only one species of lichens occurs, which**
**suggests erratic identification. The age dating of the sites is the crucial issue for the**
**study. The appearance of both higher plants native to Maritime Antarctica already on**
**the second youngest soils is directly leading to the question if the dating was done**
**correctly, since this appearance is either limited to highly developed soils or**
**ornithogenic soils. As the soil parameters suggest a young soil, not having enough N**
**for maintaining populations of higher plants on the long run, it is very likely, that these**
**encounters are due to the frequently seen bird dropping effect (excrements containing**
**seed of Deschampsia or Colobanthus stage the appearance of these plants for a season**
**or two, until the N from the excrement is used up). A coverage of 10% of higher plants**
**on such young soils suggest a high frequency of bird visits, which was excluded by the**
**authors, as this would largely question the results of a real soil development situation**
**of the microbial community, as higher plants alien to the respective community would**
**have been introduced.**

Bird droppings cannot be excluded, but we suspect the influence to be small: The P contents
in the profiles KGI A, B, and C are low and very similar for all depths. A high frequence of bird
visits including the bird droppings, as you suggest, would also lead to an increase of the P
content at least in the upper cm of these soils.  We included a table with RFA data to the
supplement (Tab. S2). Please also note, *Deschampsia antarctica* and *Colobanthus*
*quitensis* use mycorrhiozal fungi (ascomycete fungi), to "...facilitate the acquistion of organic
nitrogen as early protein breakdown products" (Hill et al. 2019). This means that they do not
need N from bird droppings to colonize new areas. Thus, the pedogenic data (particularly pH
and SOC) reflect a "real soil development" as we already have discussed in the manuscript.

**With the given data all this stays mere speculation, of course. To solve this issue, at**
**least a much better map of the sites, ideally a satellite map showing the features of the**
**sites as well as their surroundings, along with a concise description of the found**
**vegetation should be offered by the authors, please.**

We added a new figure (Figure 2) containing three satellite maps: One with very high
resolution, one with false colours to visualize the vegetation and one with the ndvi to indicate
the presence of chlorophyll. All remaining figures move one number further with their
numbering.

[Figure]

[Figure]

[Figure]

We added the caption of Figure 2:

"(A) High resolution satellite image from 06.11.2016 (Map data © 2016 Google). (B) False
colour Sentinel-2 image from 19.01.2020 (red = band 8, green = band 4, blue = band 3) for
enhanced visualisation of the vegetation. (C) Normalized Difference Vegetation Index (NDVI), dimensionless with positive values indicating healthy vegetation and negative values indicating
the presence of open water bodies. Contains modified Copernicus data." (L. 725 - 728)

Additionally, we added some information in the Materials and Methods section:

"Figure 2B, which shows increasing plant-coverage with an increasing red coloration, shows
no vegetation at KGI A, little vegetation at KGI B and C and a higher vegetation coverage at
KGI D. Conversely, image 2C shows highest values of chlorophyll at KGI D, substantially
less chlorophyll at KGI B and C and no chlorophyll at KGI A."

Regarding the description of the vegetation; please see our answer to your comment above.

**If possible some proof of the assumed age, for instance by a combination by 14C and**
**15N analyses of the soils, showing the age of organic carbon and the source of N (fixed**
**from air or carried in by birds?) would help explain this very rare combination of**
**observed features reported here. These aspects should be discussed in detail, too.**

Thank you for the suggestions.

Please note our answer to the first comments regarding age determination.

As explained in the answer to an above comment, due to similar P contents at all depths of
profiles A, B, C it is assumed that bird droppings are not of major importance at these sites
and therefore no $^{15}$N analyses were carried out.

**Regarding the DNA-based identification of bacteria it stays largely unclear to the reader**
**where the usually pretty high number of unidentified OTUs (so to say the bacteria not**
**yet known to science) in soils of Maritime Antarctica has gone. The abundance graph**
**reads as if the authors could attribute every single OTU to a phylum.**

Originally, completely unassigned OTUs have been identified, but were very low in their
abundance and summarized with other low abundant phyla under the term "Others" in Figure
3.

As suggested by Reviewer 1, the sequencing data was reanalysed by using the amplicon
sequence variant (ASV) approach. Similar to the OTU analysis, this approach resulted in low
abundances of completely unassigned sequences (minimum: 0.01 % in KGI A 10 – 20 cm;
maximum: 0.28% in KGI A 0 – 1 cm). Low abundances of unassigned reads are common in
recent literature (0.28 % of the total data set of Meier et al., 2019; 1.1 % of the total data set in
Kim et al., 2019). The very low abundance of unassigned sequences does not mean that an
identification to genus/species level for the remaining sequences was possible, though. Many
of the OTUs/ASVs shown in Fig. 5, which represent the most abundant reads in the data set,
were just classified to the order or family level.

Since both reviews wondered about this very topic, two tables were added to the supplement
as suggested by Reviewer 1: Table S4 shows the "fate" of the raw data, and Table S5 shows
the resulting ASV table. This time, both tables also include the positive as well as the negative
control. We hope the increased transparency is helpful for the evaluation of our sequencing
data.

**In the discussion, as already indicated by the missing hypotheses, the reader misses a**
**true discussion of the "identification of processes" promised in the objectives. Not even**
**the "influences of microorganisms on soil formation" promised by the title were**
**discussed in detail, what anyway would have been impossible by a study leaving the**
**fungi completely aside.**

Thank you. Please note our previous comment on the hypothesis.

To account for the missing fungi, we propose to change the title to "Influence of prokaryotic
microorganisms on initial soil formation along a glacier forefield on King George Island,
maritime Antarctica".

**After what is discussed in the paper, this study is more investigating the influence of**
**vegetation on the bacterial community of young soils in Maritime Antarctica. A big step**
**towards the originally intended process elucidation of soil formation would be for**
**instance a detailed discussion on the functional traits and abundances of bacteria**
**involved in e.g. C and N accumulation or weathering processes. In the current form, the**
**study is a family list of bacteria of four Antarctic soils, trying to mimic process**
**identification by naming rather weak coincidences. There is surely more to it.**

Thank you for your suggestions.

Since we have not included metagenomics or metatranscriptomics in our study, only limited
conclusions can be drawn about the occurrence or frequency of different functional
characteristics. On the other hand, we discuss the functions and potential effects on the soil
environment based on the taxonomy of certain observed taxa, e.g. microorganisms potentially
linked to weathering processes (L. 394 – 397) or the degradation of organic compounds (L.
265 – 269 ; L. 410 – 414). However, we agree that such aspects can be discussed in more
detail, especially after reanalysis of our sequencing data using the amplicon sequence variant
approach. Therefore, we broadened the discussion and included more information on the
potential involvement of different prokaryotic groups in certain aspects of soil formation and
rephrased the general statement on the observed composition of the community on a phylum
level:

"In accordance with observations in other Antarctic habitats (e.g. Ganzert et al., 2011, Yan et
al., 2017; Meier et al, 2019), the investigated soils were characterized by highly diverse
microbial communities including Acidobacteriota, Bacteroidota, Verrucomicrobiota and
especially high abundances of Actinobacteriota and Proteobacteria, which are known to thrive
in recently deglaciated soils and facilitate a multitude of different phototrophic,
photoheterotrophic and chemolithotrophic processes (Rime et al., 2016; Wei et al., 2016;
Garrido-Benavent et al., 2020).

Additionally, we modified the part discussing the missing indication of "obvious" prokaryotic
organisms associated with phototrophic carbon fixation:

"However, our dataset showed only low abundances of prokaryotic organisms most probably
associated with phototrophic carbon fixation, such as the Cyanobacteria-related Tychonema.
Low abundances of Cyanobacteria in recently deglaciated areas are not an uncommon
observation in Antarctic soil environments (Ji et al., 2016; Garrido-Benavent et al., 2020).In
addition to the low abundances of Cyanobacteria, other bacterial groups might be involved in phototrophic carbon fixation, such as Chloroflexi (Lacap et al., 2011). Moreover, based on the
on the amount of chloroplast-related sequences in our dataset, this process is at least partly
facilitated by eukaryotic organisms such as algae in the early stage of soil development." (L.
288ff)

We also added additional information on the influence of microorganisms on weathering
processes:

"By performing enzymatically catalysed reactions, processes reducing the pH and the
production of complexing agents (Mavris et al., 2010; Styriakova et al., 2012; Ahmed and
Holmström, 2015) microorganisms are able to substantially promote weathering processes
(Schulz et al., 2013)." (L. 391)

As mentioned above, microorganisms can facilitate processes that lower the local soil pH,
thereby promoting weathering processes. In the absence of vegetation, we observed a
decreased soil pH in the upper centimetres of the KGI A soil profile, which can therefore be
linked to microbial metabolism (L. 412 – 414). This pH reduction pH should increase as soon
as the soils are colonized by plants, since microorganisms are able to degrade organic
compounds and organic acids could be released into the surrounding soil (L. 267 – 269). This
effect can be seen in the pH reduction in relation to increasing plant coverage from KGI A (pH
6.8), KGI B (pH 6.6) to KGI C with a pH of 5.4 in within the first centimetre.

Table S1: Radiocarbon dating results obtained on the humin fraction of soil organic matter from two sites at the Ecology Glacier, King George Island.

| Profile | Depth [cm] | Horizon | Material | pMC [%] | δ13C (‰) | cal BP (1σ) | cal CE (1σ) | Lab-Nr. |
|---|---|---|---|---|---|---|---|---|
| KGI C | 0 – 1 | | soil organic matter, humin fraction (alkali soluble organics) | 112.55 ± 0.42 | -26.1 | -44 to -45 cal BP | 1993 - 1994 | Beta-570459 |
| KGI D | 0 - 3 | Ah | soil organic matter, humin fraction (alkali soluble organics) | 100.62 ± 0.38 | -27.9 | -5 to -6 cal BP | 1954 - 1955 | Beta-570458 |

Table S2: Major elements by XRF of four soil profiles from King Georges Island, Antarctica. All data given in weight percent. LOI (loss on ignition) determined at 1000°C. For location of the profiles, see table 1.

| Profile | Depth [cm] | $SiO_2$ | $Al_2O_3$ | $Fe_2O_3$ | MnO | MgO | CaO | $Na_2O$ | $K_2O$ | $TiO_2$ | $P_2O_5$ | LOI | Sum |
|---|---|---|---|---|---|---|---|---|---|---|---|---|---|
| | | | | | | | % | | | | | | |
| KGI A | | | | | | | | | | | | | |
| | 0-1 | 49,43 | 0,93 | 19,39 | 8,53 | 0,20 | 3,38 | 5,85 | 3,13 | 1,16 | 0,24 | 7,46 | 99,85 |
| | 1-10 | 48,98 | 0,94 | 19,30 | 8,48 | 0,20 | 3,15 | 5,72 | 3,00 | 1,18 | 0,25 | 8,39 | 99,73 |
| | 10-20 | 49,25 | 0,91 | 19,26 | 8,26 | 0,19 | 3,03 | 5,95 | 3,04 | 1,19 | 0,25 | 8,30 | 99,78 |
| | 20-40 | 49,73 | 0,93 | 19,12 | 8,54 | 0,20 | 3,11 | 6,26 | 3,41 | 1,16 | 0,24 | 7,12 | 99,96 |
| KGI B | | | | | | | | | | | | | |
| | 0-1 | 49,80 | 0,95 | 18,77 | 8,72 | 0,21 | 3,26 | 6,31 | 3,74 | 1,14 | 0,27 | 6,28 | 99,59 |
| | 1-10 | 49,72 | 0,94 | 18,95 | 8,71 | 0,21 | 3,29 | 6,29 | 3,65 | 1,12 | 0,26 | 6,87 | 100,14 |
| | 10-20 | 49,92 | 0,93 | 18,85 | 8,69 | 0,21 | 3,18 | 6,34 | 3,76 | 1,15 | 0,27 | 6,36 | 99,81 |
| | 20-80 | 49,66 | 0,94 | 18,83 | 8,73 | 0,20 | 3,25 | 6,51 | 3,71 | 1,14 | 0,26 | 6,43 | 99,83 |
| KGI C | | | | | | | | | | | | | |
| | 0-1 | 48,43 | 0,85 | 18,65 | 8,30 | 0,20 | 3,47 | 6,23 | 3,62 | 1,05 | 0,27 | 8,73 | 99,94 |
| | 1-10 | 49,45 | 0,94 | 18,91 | 8,76 | 0,20 | 3,22 | 6,24 | 3,63 | 1,09 | 0,28 | 6,88 | 99,75 |
| | 10-20 | 49,66 | 0,95 | 18,91 | 8,89 | 0,21 | 3,09 | 6,29 | 3,69 | 1,13 | 0,29 | 6,68 | 99,94 |
| | 20-40 | 49,73 | 0,94 | 18,90 | 8,67 | 0,19 | 3,10 | 6,30 | 3,73 | 1,17 | 0,29 | 6,35 | 99,52 |
| KGI D | 0-3 | 44,39 | 0,79 | 17,47 | 7,75 | 0,17 | 3,40 | 5,88 | 2,98 | 1,12 | 1,04 | 14,6 | 99,76 |
| | 3-15 | 48,84 | 0,82 | 19,22 | 8,06 | 0,18 | 3,54 | 6,30 | 3,53 | 1,19 | 0,35 | 7,84 | 100,01 |
| | 15-27 | 49,85 | 0,80 | 19,28 | 7,81 | 0,19 | 3,50 | 6,80 | 3,79 | 1,15 | 0,21 | 6,54 | 100,05 |
| | 27-60 | 49,24 | 0,82 | 19,30 | 8,04 | 0,20 | 3,56 | 7,01 | 3,71 | 1,08 | 0,19 | 6,49 | 99,80 |

Table S4: Number of sequencing reads after each processing step.

| sample | input | filtered | denoised | merged | Non-chimera | 0.01% cutoff |
|---|---|---|---|---|---|---|
| | | | number of reads | | | |
| KGI_A_0_1_a | 823555 | 760184 | 744842 | 728869 | 722825 | 706168 |
| KGI_A_0_1_b | 416652 | 386897 | 378089 | 368092 | 365389 | 359124 |
| KGI_A_0_1_c | 923881 | 848435 | 824280 | 799372 | 786254 | 772324 |
| KGI_A_1_10_a | 473836 | 436670 | 427179 | 416676 | 413400 | 410050 |
| KGI_A_1_10_b | 234950 | 218204 | 212241 | 204825 | 203788 | 201502 |
| KGI_A_1_10_c | 352840 | 330275 | 323230 | 316343 | 314148 | 311452 |
| KGI_A_10-20_a | 36923 | 33865 | 29718 | 26130 | 25958 | 25922 |
| KGI_A_10-20_b | 18966 | 17275 | 14209 | 12268 | 12259 | 12252 |
| KGI_A_10-20_c | 25875 | 23729 | 19871 | 17313 | 17308 | 17300 |
| KGI_A_20_40_a | 905510 | 839764 | 828694 | 814373 | 808267 | 790148 |
| KGI_A_20_40_b | 846168 | 790425 | 779458 | 764252 | 758482 | 737591 |

| | | | | | | |
|---|---|---|---|---|---|---|
| KGI_A_20_40_c | 893543 | 830439 | 821643 | 808578 | 801514 | 778446 |
| KGI_B_0_1_a | 802491 | 738909 | 726606 | 711994 | 705038 | 683957 |
| KGI_B_0_1_b | 767042 | 707072 | 685728 | 660207 | 650385 | 634947 |
| KGI_B_0_1_c | 776140 | 717207 | 691876 | 662511 | 644483 | 630913 |
| KGI_B_1-10_a | 870220 | 801229 | 775510 | 742168 | 726536 | 700009 |
| KGI_B_1-10_b | 763113 | 708362 | 698110 | 683584 | 679585 | 655095 |
| KGI_B_1-10_c | 984339 | 904070 | 867753 | 821445 | 790478 | 763807 |
| KGI_B_10_20_a | 902645 | 835484 | 829409 | 818332 | 809608 | 796113 |
| KGI_B_10_20_b | 50853 | 47043 | 41987 | 37904 | 37729 | 37613 |
| KGI_B_10_20_c | 45144 | 41598 | 36756 | 32746 | 32618 | 32557 |
| KGI_B_20_80_a | 797077 | 739614 | 734477 | 727607 | 713996 | 706767 |
| KGI_B_20_80_b | 666167 | 617867 | 613646 | 605338 | 594001 | 587915 |
| KGI_B_20_80_c | 59859 | 55236 | 50726 | 47120 | 46730 | 46649 |
| KGI_C_0_1_a | 748549 | 692351 | 673800 | 648601 | 635472 | 605915 |
| KGI_C_0_1_b | 896152 | 828214 | 784842 | 728938 | 687678 | 661043 |
| KGI_C_0_1_c | 780292 | 724237 | 701902 | 672607 | 650189 | 619970 |
| KGI_C_1_10_a | 1072129 | 979738 | 941569 | 893583 | 866371 | 830430 |
| KGI_C_1_10_b | 796453 | 734715 | 714128 | 686413 | 675197 | 651021 |
| KGI_C_1_10_c | 945868 | 879402 | 852158 | 818052 | 799307 | 765149 |
| KGI_C_10_20_a | 914463 | 847130 | 837834 | 821943 | 814309 | 788037 |
| KGI_C_10_20_b | 724047 | 663724 | 657595 | 646501 | 642849 | 625122 |
| KGI_C_10_20_c | 379777 | 348578 | 342785 | 334296 | 331763 | 325740 |
| KGI_C_20_40_a | 805325 | 743983 | 737314 | 727505 | 720592 | 705527 |
| KGI_C_20_40_b | 883724 | 818626 | 808175 | 789997 | 782377 | 762057 |
| KGI_C_20_40_c | 651443 | 603352 | 595920 | 583664 | 578353 | 564023 |
| KGI_D_0_3_a | 779880 | 715854 | 698512 | 675201 | 664504 | 626027 |
| KGI_D_0_3_b | 787246 | 723529 | 685098 | 643359 | 617163 | 586001 |
| KGI_D_0_3_c | 898392 | 826444 | 770393 | 701924 | 643856 | 619137 |
| KGI_D_3_15_a | 815335 | 754410 | 739659 | 718535 | 709893 | 677501 |
| KGI_D_3_15_b | 950935 | 871121 | 850341 | 820666 | 806893 | 766920 |
| KGI_D_3_15_c | 930601 | 858951 | 806053 | 732072 | 650658 | 627047 |
| KGI_D_15_27_a | 913147 | 851119 | 839448 | 822669 | 812036 | 784157 |
| KGI_D_15_27_b | 1074041 | 989909 | 980950 | 964946 | 953467 | 917368 |
| KGI_D_15_27_c | 869976 | 799507 | 791364 | 777160 | 770119 | 749250 |
| KGI_D_27_60_a | 740686 | 687100 | 681491 | 671687 | 666209 | 645648 |
| KGI_D_27_60_b | 768476 | 714698 | 707202 | 695134 | 689840 | 671215 |
| KGI_D_27_60_c | 852927 | 790621 | 782334 | 769492 | 763600 | 737767 |
| Negative control | 11045 | 9999 | 9955 | 9899 | 9899 | 9819 |
| Positive control | 651120 | 590140 | 589778 | 585268 | 585268 | 585250 |

References

Ahmed, E., and Holmström, S. J.: Microbe–mineral interactions: the impact of surface attachment on mineral weathering and element selectivity by microorganisms. Chemical Geology, 403, 13-23, 2015.

Garrido-Benavent, I., Pérez-Ortega, S., Durán, J., Ascaso, C., Pointing, S. B., Rodríguez-Cielos, R., Navarro, F. and de los Ríos, A.: Differential Colonization and Succession of Microbial Communities in Rock and Soil Substrates on a Maritime Antarctic Glacier Forefield. Frontiers in microbiology, 11, 126, 2020.

Ganzert, L., Lipski, A., Hubberten,H.-W., and Wagner, D.: The impact of different soil parameters on the community structure of dominant bacteria from nine different soils located on Livingston Island, South Shetland Archipelago, Antarctica, FEMS microbiology ecology, 76, 476–491, 10.1111/j.1574-6941.2011.01068.x, 2011

Hill, P.W., Broughton, R., Bougoure, J., Havelange, W., Newsham, K.K., Grant, H., Murphy, D.V., Clode, P., Ramayah, S., Marsden, K.A., Quilliam, R.S., Roberts, P., Brown, C., Read, D.J., Deluca, T.H., Bardgett, R.D., Hopkins, D.W. and Jones, D.L.: Angiosperm symbioses with non-mycorrhizal fungal partners enhance N acquisition from ancient organic matter in a warming maritime Antarctic, Ecology Letters, 22: 2111-2119, 2019.

Ji, M., van Dorst, J., Bissett, A., Brown, M. V., Palmer, A. S., Snape, I., Siciliano, S. D. and Ferrari, B. C.: Microbial diversity at Mitchell Peninsula, Eastern Antarctica: a potential biodiversity "hotspot". Polar Biology, 39(2), 237-249, 2016.

Kim, M., Lim, H. S., Hyun, C. U., Cho, A., Noh, H. J., Hong, S. G., and Kim, O. S.: Local-scale variation of soil bacterial communities in ice-free regions of maritime Antarctica, Soil Biology and Biochemistry, 133, 165-173, 2019.

Lacap, D. C., Warren-Rhodes, K. A., McKay, C. P., and Pointing, S. B.: Cyanobacteria and chloroflexi-dominated hypolithic colonization of quartz at the hyper-arid core of the Atacama Desert, Chile. Extremophiles, 15(1), 31-38, 2011.

Mavris, C., Egli, M., Plötze, M., Blum, J. D., Mirabella, A., Giaccai, D., and Haeberli, W.: Initial stages of weathering and soil formation in the Morteratsch proglacial area (Upper Engadine, Switzerland). Geoderma, 155(3-4), 359-371, 2010.

Meier, L. A., Krauze, P., Prater, I., Horn, F., Schaefer, C. E. G. R., Scholten, T., Wagner, D., Mueller, C. W., and Kühn, P.: Pedogenic and microbial interrelation in initial soils under semiarid climate on James Ross Island, Antarctic Peninsula region, Biogeosciences, 16, 2481–2499, 10.5194/bg-16-2481-2019, 2019.

Rime, T., Hartmann, M., Stierli, B., Anesio, A. M., and Frey, B.: Assimilation of microbial and plant carbon by active prokaryotic and fungal populations in glacial forefields. Soil Biology and Biochemistry, 98, 30-41, 2016

Schulz, S., Brankatschk, R., Dümig, A., Kögel-Knabner, I., Schloter, M., and Zeyer, J.: The role of microorganisms at different stages of ecosystem development for soil formation. Biogeosciences, 10(6), 3983-3996, 2013.

Styriakova, I., Styriak, I., and Oberhänsli, H.: Rock weathering by indigenous heterotrophic bacteria of Bacillus spp. at different temperature: a laboratory experiment, Mineral. Petrol., 105, 135–144, 2012.

Wei, S. T., Lacap-Bugler, D. C., Lau, M. C., Caruso, T., Rao, S., de Los Rios, A., Archer, S.
K., Chiu, J. M., Higgins, C., Van Nostrand, J. D., Zhou, J., Hopkins, D. W. and Pointing, S. B.:
Taxonomic and functional diversity of soil and hypolithic microbial communities in Miers Valley,
McMurdo Dry Valleys, Antarctica. Frontiers in microbiology, 7, 1642, 2016.
Yan, W., Ma, H., Shi, G., Li, Y., Sun, B., Xiao, X., and Zhang, Y.: Independent Shifts
of Abundant and Rare Bacterial Populations across East Antarctica Glacial Foreland,
Frontiers in microbiology, 8, 1534, 10.3389/fmicb.2017.01534, 2017.